# Determination of Acoustic Compliance of Wind Farms

**Steven Cooper \* and Christopher Chan** 

The Acoustic Group, South Windsor 2756, Australia; chris@acoustics.com.au
\* Correspondence: drnoise@acoustics.com.au; Tel.: +612-9555-4444

**Abstract:** An issue exists around the world of wind farms that comply with permit conditions giving rise to noise complaints. Approval limits are normally expressed in A-weighted levels (dB(A)) external to residential receivers. The distance from the wind farm to residential receivers can result in difficulty in establishing the dB(A) contribution of the wind farm, as the overall noise includes background noise that can provide masking of the wind turbine noise. The determination of the ambient background at a receiver location (without the influence of the wind farm) presents challenges, as the background level varies with the wind and different seasons throughout the year. On-off testing of wind farms does not normally occur at high wind farm output and limits this approach for acoustic compliance testing of a wind farm. The use of a regression analysis method developed more than 20 years ago is questioned. Anomalies with respect to compliance procedures and the regression method of analysis based on real-world experience are discussed.

**Keywords:** wind turbine criteria; wind noise; ambient background level; regression analysis

## 1. Introduction

The acoustic parameter in common use for the assessment of wind turbines is the dB(A) level, assessed as an energy averaged level ($L_{Aeq}$). Some countries use a metric that is derived from the $L_{Aeq}$ metric that include a penalty for time of the day related to an increased sensitivity for evening and night [1,2]. Noise limits for wind turbines relate to the noise contribution attributed to the wind turbine and does not relate to the overall noise level that includes ambient noise from the surrounding environment and noise from the wind.

As the operation of wind turbines requires wind, it is necessary to determine the $L_{Aeq}$ contribution of the wind farm, to exclude the underlying ambient background level (that includes noise from the wind), noting that the ambient background level as a result of the wind will increase as the wind speed increases.

Around the world, there are a wide range of noise regulations governing wind turbine noise. The most common noise metric is based on the equal energy level ($L_{eq}$). Depending on the variation of the $L_{eq}$ that is used and the classification of the land use in proximity to a wind farm, the range of noise limits can exhibit significant differences.

In Sweden [3] and Denmark [1], there are $L_{eq}$ limits for a reference wind speed of 8 m/s at a height of 10 m above ground level. In Netherlands there is a day/evening/night limit ($L_{den}$) and a night limit ($L_{night}$). France [4], Belgium [5] and Norway [6] use an $L_{Aeq}$ limit. Most countries in Europe use fixed noise limits (that are not the same) for different time periods. France references the limit to the background level, whilst Germany [7] uses a noise rating level ($L_r$) based upon the German DIN Standard 45645-1 [8], that measures the $L_{eq}$ level and applies penalty for possible tonal noise.

In 2018 the WHO included noise criteria for wind turbines (Environmental Noise Guidelines for the European Region) [9]. With the qualification of limited data in relation to annoyance (four studies), a conditional limit of 45 $L_{den}$ (outdoors) was recommended for an average noise exposure.

There is no national legislation for wind turbine noise in Canada. In Ontario (Canada) there are different $L_{eq}$ limits for urban (Class 1 and 2) and rural (Class 3) $L_{eq}$ limits. The Ontario limits [10] adopt a base level re wind speed at 10 m AGL, to a set integer wind speed from which the limit increases. For Class 3 (rural) areas, the lowest value is 40 dB(A) at wind speeds at or below 6 m/s, to a maximum of 51 dB(A) for wind speeds at or above 10 m/s. For urban areas, the lowest limit is 45 dB(A), at or below 8 m/s wind speed.

A method developed in the UK [11] for the assessment of wind farms (ETSU-R-97) adopted the general assessment approach of $L_{A90}$ background +5 dB(A) as a starting point, but fixed absolute lower noise limits. ETSU-R-97 nominated the use of a regression analysis method to derive an average background level versus wind speed. Compliance testing utilizes the regression analysis method with the wind farm operating, to ascertain the wind farm contribution.

In Australia and New Zealand, ETSU-R-97 has been adopted as the base assessment method of assessment. The wind turbine noise limits in Australia and New Zealand have been strongly influenced by ETSU-R-97 and adopted a background +5 dB(A) approach, with fixed absolute lower noise limits [12].

The above criteria recommend measuring and assessing the outdoor sound levels at external receiver locations rather than inside dwellings.

In the US, there is no national legislation for wind turbine noise. The majority of states use an $L_{Aeq}$ metric and a few states use an $L_{A10}$ metric. Individual states in the US may use fixed limits of between 45–55 dB(A), background +5 dB(A), or a fixed limit and background +5 dB(A) [13–21]. This leads to a range of noise limits in the US that are generally higher than the noise limits used in Europe, the UK and Australasia. Such higher noise limits (in the US) can result in residential receivers closer to the wind farm than for the Swedish, Dutch, Canadian and Australian examples provided above.

Whilst noise limits may be expressed in $L_{eq}$ levels, the ETSU regression analysis method uses the $L_{90}$ A-weighted value for addressing noise impacts experienced by the community. The relevance of the noise signature from wind turbines may not necessarily be identified by the $L_{90}$ parameter.

The issue of health impacts as a result of noise from wind farms and the appropriate noise limits to protect against health impacts is unknown. The Massachusetts Wind Turbine Health Study [22] identifies the need for wind turbine sleep studies to determine health impacts, whilst Figure 4.3 in the WHO Night Guidelines [23] identify that ongoing sleep disturbance can result in health impacts.

The WHO 2018 Guidelines (for the European Region [9]) noted the low quantity and heterogeneous nature of the evidence on sleep disturbance. There is no nighttime criteria specified for wind turbine noise to protect against sleep effects in the WHO 2018 Guidelines.

The derivation of an accurate noise contribution of a wind farm is important for assessing health and noise impacts and deriving noise dose curves in relation to such impacts. The main aim of this technical note is to identify issues related to the difficulty in obtaining a dB(A) contribution from a wind farm when examined in light of real-world measurements.

## 2. The Development of Wind Turbines for Electric Power

In 1973, a national wind energy program was established in the US, to develop wind turbines for electric power, in response to an increase in oil prices [24]. The National Aeronautics and Space Agency (NASA) managed the project to develop utility-scale wind turbines and transfer the research and technology from the Government to the commercial sector. Subsequently, various companies around the world developed large scale wind turbines that could connect to the electricity grid [25].

NASA developed a number of horizontal axis 2 bladed downwind turbines commencing with first generation machines (Mod-0 and Mod-0A), involving a 100 kW and a 200 kW unit, respectively. The Mod-0 turbines were not considered to create significant noise [26].

The Mod-1 project was initiated in 1974 based on the Mod-0 results and involved the construction of a 200 ft diameter wind turbine with a rated power of 2000 kW. At the time, the turbine represented

the largest in the world. The Mod-0 and Mod-1 turbines were developed with the turbine blades being downwind of the tower structure, which was a lattice type construction.

The Mod-1 turbine resulted in noise complaints to a small fraction of families living within a 3 km radius of the turbine. A detailed investigation was undertaken by the Solar Energy Research Institute ("SERI") for the US Department of Energy [27].

Kelley et al. [27] identified the acoustic pulsations emitted from the Mod-1 turbine, the level of sound above human perception thresholds and the excitation of building/room modes in dwellings.

The Executive Summary of the SERI report [28] identifies the perception of the complaints for the single Mod-1 turbine are similar to that obtained for current wind farm (multiple wind turbine) installations. The key findings in the SERI report were that the annoyance was real and not imagined, and the source of the annoyance was aerodynamic, that involved the passage of the turbine blades through the lee wakes of the large, 0.5 m cylindrical tower legs. In some instances, the acoustic impulses transmitted through the air were being focused on the complainant's homes, as a consequence of ground reflection and refraction by the atmosphere.

As a result of the Mod-1 acoustic issues, the design of the NASA turbine program moved to the use of upwind turbines, starting with Mod-2, where the turbine blades were positioned in front of a circular tower.

Hubbard and Shepherd [29] identify the significant reduction in the impulsive characteristics achieved by an upwind rotor, to that of a downwind rotor turbine from the NASA program (see Figures 7-4 and 7-5 in reference [29]). In an acoustic sense the results of the Mod-2 2500 kW upwind turbine were significant, resulting in a conclusion that annoyance to the community was very unlikely at distances greater than 1 km [30].

In light of the low frequency sound from the Mod-1 turbine investigations, Kelley examined several acoustic metrics for assessing the interior low frequency annoyance [31], taking into account the overall degree of annoyance, any sensations of vibration or pressure or the sensing of any pulsations. Kelley proposed the use of the LSL or low frequency sound level weighting [32] or the C-weighting. Kelley's proposal was not adopted by authorities, although Jakobsen [33] proposed an internal low frequency parameter $L_{pA, LF}$ as the A-weighted level, in the frequency range of 10-160 Hz. The Jakobsen limit is used in Denmark.

## 3. Acoustic Criteria

The noise targets nominated for wind turbine installations rely upon criteria based upon general environmental noise policies, that are focused upon the A-weighted metric and involve fixed limits and/or a background +5 dB(A) concept.

Noise limits used in Australasia [34–39] and the UK [11] identify that the acoustic criteria is to protect people from sleep disturbance. The WHO sleep criteria of 30 and 35 dB(A) indoors (in the 1990s) for community noise [40,41] is generally used as a reference for sleep disturbance, but without qualifying that the criteria was based upon transportation noise, primarily road traffic noise. Where the internal noise contains a low frequency component, then a further adjustment to the limit is suggested.

Pedersen, Van den Berg, Bakker and Bourma [42], in reporting on the response to noise from modern wind farms in the Netherlands, provided (in 2009) a comparison of dose-response curves of wind turbine noise with other sources of community noise. The dose response curves developed by Pedersen et al. had a small number of data points, but showed that "the proportion of respondents annoyed with wind turbine noise below 50 dB(A) $L_{den}$ is larger than the proportion annoyed with noise from all other sources except shunting yards".

A relevant issue to the curves that were derived was that the material relied upon predicted wind turbine noise. The Pedersen et al. paper does not identify any correlation of predicted levels versus measured levels, or the actual operation of the wind farms relative to the assumed operation used for the predicted levels.

Janssen, Vos, Eisses and Pedersen [43] reviewed the data from the three studies in reference [42], to derive a dose response curve for wind turbines that is included in the WHO 2018 Guidelines [9] in relation to wind turbines. The WHO Guidelines provide in the section dealing with wind turbine noise a dose response curve from Kuwano [44] that does not agree with the Janssen dose response curves.

Figure 1 presents the three dose response curves for wind turbine noise referred to above as $L_{den}$ values, and the $L_{den}$ 45 limit for wind turbines and the WHO $L_{den}$ limit. Included in Figure 1 (for comparison purposes) is the WHO road traffic dose response curve (extracted from the 2018 WHO Guideline [9]), noting that the WHO recommend a 53 $L_{den}$ limit for adverse health effects and related to 10% of the population being highly annoyed.

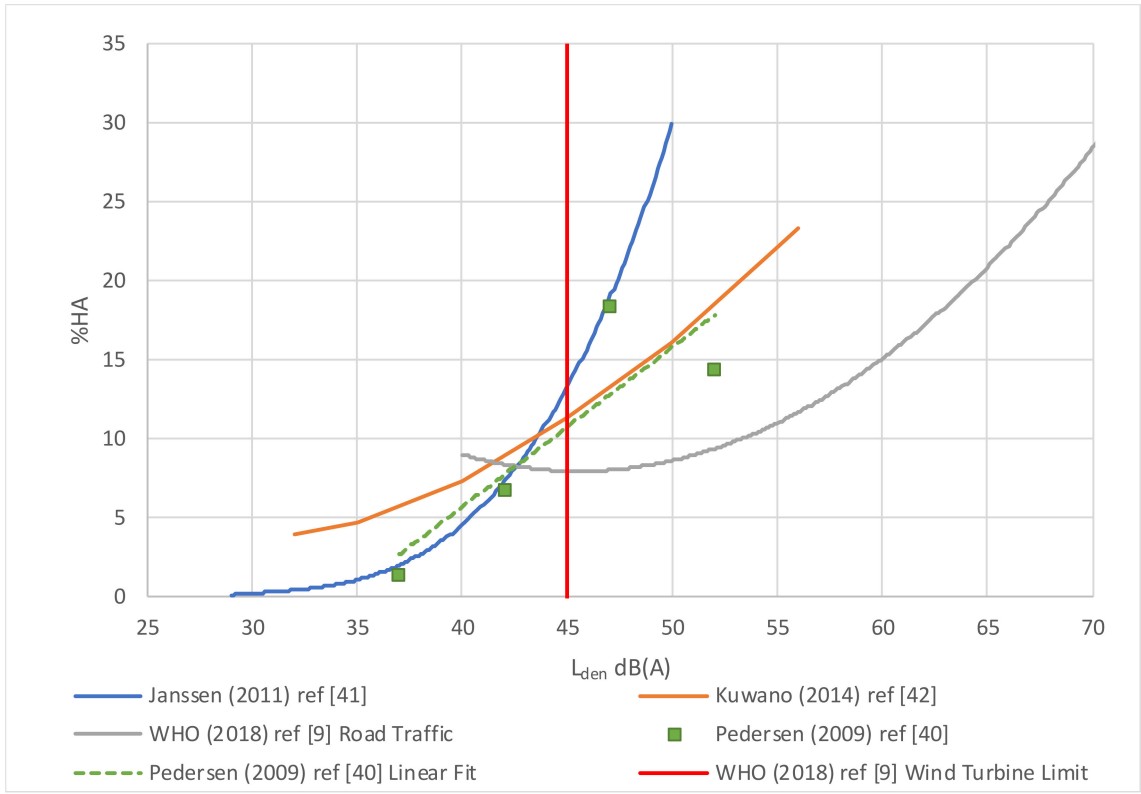

**Figure 1.** Proportion of respondents highly annoyed from wind turbines, Pedersen (2009), Janssen (2011), Kuwano (2014), re $L_{den}$ and compared with road traffic (WHO 2018).

The Kuwano curve in Figure 1 relies upon measurements near the wind farm and an extrapolation to the surveyed areas. The Kuwano curve in the WHO Noise Guidelines (2018) was presented as an $L_{dn}$. On the assumption that the Kuwano wind farm noise is constant throughout the day, the Kuwano curve in Figure 1 has been corrected to an $L_{den}$.

Adopting the general approach of setting a noise limit and the basis of 10% of the population being highly annoyed, from Figure 1, the WHO limit should be $L_{den}$ 43.5 dB(A).

On the assumption of a steady noise level throughout a 24 h period to convert an $L_{den}$ to an $L_{eq}$ one subtracts 6.7 dB(A) off the $L_{den}$ value. Figure 2 presents the results in Figure 1 to $L_{eq}$ levels, to permit a comparison with Leq limits used in other countries around the world.

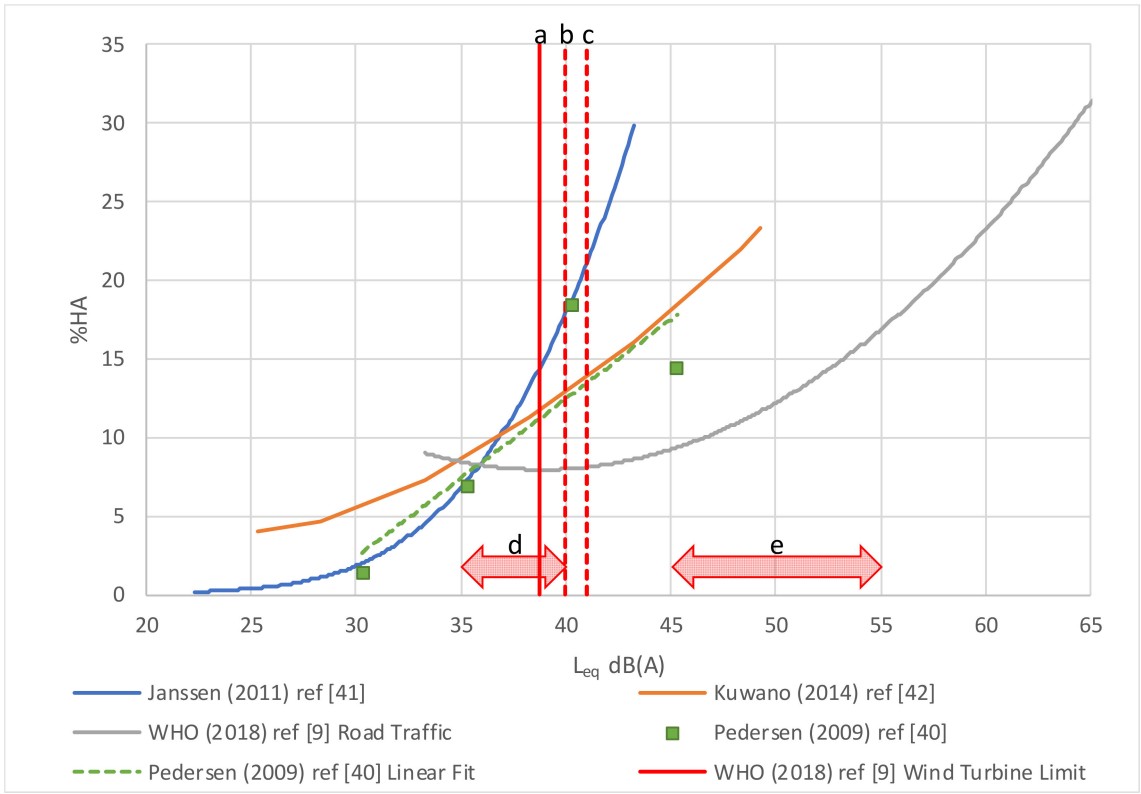

**Figure 2.** Proportion of respondents highly annoyed from wind turbines, Pedersen (2009), Janssen (2011), Kuwano (2014), re $L_{den}$ and compared with road traffic (WHO 2018) in terms of Leq levels. N.B. **a.** is the WHO wind turbine noise limit, **b.** is the WHO road traffic limit, **c.** is the Norway night time limit, **d.** is the minimum limits in Australia and **e.** is the range of limits in the USA.

In Figures 1 and 2, it can be seen that, if the acoustic impact of wind turbines is greater than the road traffic noise, then the basis of wind turbine noise targets that were determined in ETSU-R-97 to protect sleep (using the WHO recommendation) could be inadequate. In the absence of the validation of the wind turbine contributions, that were impacting upon the surveyed population, the actual contributions could be higher or lower than the predicted levels.

A similar result to the above surveys is obtained in relation to the Canadian Health study into two wind farms [45,46]. The analysis of the noise results with respect to impacts in the Canadian Health Study were based upon predicted levels [47], with no verification of the wind turbine noise levels experienced by the people involved in the study. Davy et al. [12] compared the percentage of highly annoyed people from the Canadian Study to the Dutch and Sweden survey data [42], to reveal a greater disturbance. Arising from the Canadian Health Study, Michaud et al. [46] reviewed published data on annoyance with wind turbines, using the community tolerance level (CTL), previously used for transportation noise [48,49].

The Cape Bridgewater study undertaken (by the authors) in Australia [50] was required to independently assess the impacts from residents' complaints—akin to a Soundscape investigation. The Cape Bridgewater Study had the unique situation of full and unfettered access to the wind farm, the provision of individual wind turbine data for the entire testing and the unique situation of multiple shutdowns of the entire wind farm (for high voltage cabling work at the main substation), that permitted the identification of the ambient noise levels in the presence of wind and no turbines operating.

The Cape Bridgewater study could not find any correlation between the operation of the wind farm and the A-weighted level (or 22 other noise metrics that were assessed). However, a correlation of the dB(A) level versus the wind speed was obtained.

As a result of investigations into residents' complaints of noise disturbance from wind farms in Australia, a number of case studies, and attempts to ascertain acoustic compliance of wind farms in Australia, led to the question: What is the dB(A) contribution from wind farms?

Where the noise limits are expressed relative to the background level, noting that the background level can increase as a result of the wind speed requires a review of the basis of the regression analysis method.

## 3.1. ETSU-R-97

The genesis of acoustic criteria and compliance testing specified for wind farms in the UK and Australia is often referenced back to ETSU–R–97 (The Assessment and Rating of NOISE from Wind Farm) [11]. In 1993, the UK Department of Trade and Industry established a Noise Working Group leading to the issue of ETSU-R-97 in 1996, based on relatively small turbines (up to 450 kW), and upon general environmental acoustic standards/policies used in the UK [51,52].

ETSU-R-97 was primarily a planning document and did not address the acoustic impact of wind turbines [53,54].

The noise working group proposed the use of $L_{A90,10 \, min}$, with the corresponding average wind speeds measured over the same 10 min periods and then fitting a curve to that data.

The working group concluded that the $L_{A90 \, 10 \, min}$ of the wind farm is likely to be about 1.5 to 2.5 dB(A) less than the $L_{Aeq}$ measured over the same period, and that the use of the $L_{A90 \, 10 \, min}$ descriptor for wind farm noise allows reasonable measurements without corruption from relatively loud, transitory noise events from other noise sources.

The methodology adopted in ETSU-R-97 considered 10-min average wind speeds measured at a 10 m height above ground level, with a polynomial fit to achieve the regression line with respect to the wind speed.

On the basis of 35 dB(A) sleep disturbance criteria referred to in Planning Policy Guidance Note 24 [52] (now withdrawn), an allowance of 10 dB(A) was made for attenuation through an open window (free-field to internal) and 2 dB subtracted to account to the use of $L_{A90 \, 10 \, min}$ rather than $L_{Aeq, \, 10 \, min}$. The report recommended that separate noise limits apply for daytime to that for night-time and identified the daytime limits would be derived from quiet periods of the day identified as all evenings from 6 p.m. to 11 p.m., plus Saturday afternoon from 1 p.m. to 6 p.m., plus all day Sunday from 7 a.m. to 6 p.m.

The ETSU report identifies 18 wind farms (in England and Wales) used in a survey of public reaction to noise from wind turbines (as reported to the environmental health departments as of February 1994), to utilize turbines having a rated power varying from 225 kW to 450 kW.

Similarly, the ETSU report provided a summary of complaints from wind farms showing that 13 of the 18 windfarms had summarized the number of complaints. Only 5 windfarms reported noise complaints.

ETSU-R-97 referenced the British Standard 4142 [51] and rejected the use of background noise levels below 30 dB(A). ETSU-R-97 suggested that instrumentation could not reliably provide background levels below 30 dB(A) and that BS 4142 was designed to assess the likelihood of complaints from people residing inside a building, based on measurements outside the building. The concept presented was that, for noise levels less than 35 dB(A), when measured externally, the masking level inside the property will be dominated by internal noise sources.

ETSU-R-97 recommended the regression analysis method be used for compliance purposes.

Issues Arising from the Use of ETSU-R-97

ETSU-R-97 has not been updated, despite changes in instrumentation, BS 4124, WHO sleep criteria, identification of "excessive amplitude modulation" and further research into wind turbine noise.

There has been debate amongst acousticians dealing with wind turbine noise, as to the suitability of ETSU-R-97. Bowdler [54] identifies that the document is not a method of assessing impact and failed

to follow the procedure of BS4124 -1990 that nominated exceedances above the background level that can give rise to significant impacts.

Walsh [55] refers to the changes in BS4124 since 1990. Walsh's submission (Section 9.5 in reference [55]) identifies that some members of the original noise working group on wind farm noise, which drafted ETSU-R-97, have raised matters that should be addressed when assessing wind farm noise.

Stigwood [56] also raises issues with BS4124 having been revised a number of times since the issue of ETSU-R-97, noting that a report issued by the UK Department for Environment, Food and Rural Affairs [57] confirms the inapplicability of ETSU-R-97 to nuisance assessment.

The ETSU report did not identify the actual or predicted wind farm noise levels, nor the ambient background level at the various wind farms identified in ETSU-R-97 to support the recommended criteria.

ETSU did not adopt the BS4142 concept of the ambient noise comprising the residual noise and the specific noise, and the need to distinguish between the specific noise and the residual noise.

ETSU-R–97 identifies that the majority of the turbine installations used as part of the assessment process did not give rise to noise complaints.

McKenzie [58] raises issues of the derived prevailing background noise curves and whether measurement locations are affected by noise from trees and foliage and/or other noise sources.

The authors have not had the opportunity to undertake testing on small turbines (<500 kW) that were in existence at the time that ETSU-R-97 was developed.

### 3.2. Australasian Criteria

In Australia and New Zealand, acoustic criteria for wind turbines have, until recent years, been based upon a 24-h regression analysis and strongly rely upon ETSU-R-97 [11].

### 3.2.1. New Zealand Standard 6808

New Zealand Standard 6808:1998 Acoustics—The Assessment and Measurement of Sound from Wind Turbine Generators [34], whilst being used in New Zealand, is also used in the state of Victoria (being on the south-eastern corner of Australia). NZ6808-1998 was the first document used in Australia for wind farms permits.

Clause 4.4.1 of NZ 6808–1998 identifies that the basis of the noise criteria set out in the Standard is to protect individuals from potential adverse effects of wind turbine generator. The same clause identifies that the basis of the criteria is for an accepted indoor sound level of 30-35 dB(A) $L_{eq}$, commonly used as a design level to protect against sleep disturbance.

The NZ Standard recommends, for rural receivers, a $L_{A95}$ noise level of 40 dB(A), or $L_{A95}$ background +5 dB, whichever is the greater.

The background level descriptor in the NZ Standard is based upon a regression line for $L_{A95\ 10-min}$ levels, recorded over a suggested period of 10–14 days.

Determining the background level at the receiver locations versus the 10 m wind at the wind farm adopts the methodology in ETSU–R-97.

The ambient background levels used in the regression line approach are obtained at residential receivers and are referenced to a wind level on the wind farm at 10 m above ground level.

The compliance methodology utilizes the same procedure and seeks to compare the pre-construction regression curve with a post-construction regression curve. The worked example in the NZ Standard identifies that it may be necessary to determine the "wind farm only" levels, but does not provide a procedure to derive those levels.

The New Zealand Standard makes no reference to any dose response curves applicable to sleep disturbance for noise from wind turbines and relies upon the Community Noise document [40].

The New Zealand Standard fails to identify that the 1995 Community Noise document, from which the 30–35 dB(A) internal noise level was obtained, does not include any noise assessments of wind

turbines, but relies upon studies related to disturbance from transportation noise, with the majority of the references related to road traffic noise.

The NZ Standard indicates that, in some situations, it may be necessary to consider separate criteria for night versus day, but does not identify any different approach in terms of the regression analysis methodology.

A second version of the New Zealand Standard NZS 6808:2010 Acoustics—Wind Farm Noise [37] reverted to the $L_{A90}$ level, not the $L_{A95}$. For the 30 dB(A) internal noise level, the Standard refers to Guidelines for Community Noise, World Health Organization 1999 by Burglund, Lindvall and Schwea [41].

In the 2010 version of the Standard, Clause 5.1.2 states the noise criteria are to provide a satisfactory level of protection against sleep disturbance. The Standard recommends a wind turbine sound level (outdoor noise) for sensitive locations of 40 dB(A) $L_{A90\,(10\,min)}$. The 2010 version of the Standard provides no reference material identifying sleep disturbance studies in relation to wind turbine noise.

Various permit approvals issued in the state of Victoria (in Australia) utilized the 1998 $L_{A95}$ assessment criteria, and are still applied for compliance purposes, notwithstanding the issue of a 2010 version.

In both versions of the NZ Standard, there is a requirement to undertake all the wind measurements (before and after installation of the wind farm) at the same location and height.

The 2010 version of NZ Standard nominated the wind to be obtained preferably at the height of the wind turbine hub.

In some cases, this presents a problem, in that the base background regression curve, if obtained under the 1998 version of the NZ Standard, would have wind data at 10 m AGL on the wind farm, whereas the 2010 version indicates that a preferred height is the wind turbine hub height.

The regression line obtained from 10 m AGL wind measurements versus a regression line from hub height wind measurements is different (by reason of greater wind speeds at the hub height) and therefore present issues where the wind farm operator/consultant choose to change the location of the wind measurement, during or after the original assessment—without identifying the difference or the relationship of the hub height wind levels versus the 10 m AGL wind levels.

As the operational wind farm wind levels (10 m AGL or hub height) are not placed in the public domain, then the community has no information to check acoustic compliance.

Furthermore, for the pre-installation wind testing, the site is free of turbines, which is not the situation after the turbines are built. Accordingly, the pre-installation wind data (and acoustic assessment) is not subject to wake turbulence—whereas the operating wind farm meteorology masts can be subject to wake turbulence that can make a high percentage of wind turbines experience unstable wind.

3.2.2. South Australian EPA Guidelines

The 2003 South Australian EPA Environmental Noise Guidelines: Wind Farms [36] identifies, in the Introduction:

The core objective of the guidelines is to balance the advantages of developing wind projects in this State with protecting the amenity of the surrounding community from adverse noise impacts.

The guideline referenced ETSU–R–97, New Zealand Standard 6808:1998 and The WHO 1999 Guidelines for Community Noise.

The guideline does not identify what constitutes an adverse noise impact for the community surrounding the wind farm or what levels represent an adverse impact.

For new wind farm developments, the Guideline nominated a predicted equivalent $L_{Aeq,\,10\,min}$ not to exceed 35 dB(A) or the background noise ($L_{A90,\,10\,min}$) by more than 5 dB(A), whichever is the greater, at all relevant receivers for each integer wind speed from cut-in to rated power of the wind turbine generator.

The methodology for determining the background level and the resultant criteria used for assessment purposes followed the ETSU–R–97 concept, but had the background levels assessed over the entire daytime period, rather than the sensitive times in the day, or night identified in ETSU. The Guideline suggests that there should be a minimum of 2000 pairs of synchronized background noise and wind speed measurements, for wind speeds between the cut-in speed and speed at rated power.

The concept of plotting the wind speed 10 m above AGL (at the wind farm) along the *x*-axis and the background noise along the *y*-axis was specified, with a best fit regression analysis carried out on the data. The polynomial order (from linear to third order) was to be undertaken to provide the best correlation coefficient for the fitted regression line.

There is no requirement to identify the relationship of the wind (speed and direction) at the receiver location to the wind farm wind measurement location.

ETSU-R-97, NZS 6808, SA EPA Wind Farm Guidelines and Australian Standard 4959 provide examples of the regression analysis method, by plotting the background L90, 10 m levels at a receiver location versus the wind speed on the subject wind farm (10 m AGL of hub height). In the above Standards/Guidelines, the regression curve is typically presented as a second order polynomial, where the curve tends to flatten out at the lower wind speeds to reveal the normal ambient background level used in general environmental assessments.

For compliance purposes, the same method is undertaken, to which the resultant regression line is compared with the original design criteria.

The difficulty for researchers is that the wind data (on the wind farm) is not in the public domain, and there restricts the ability to undertake independent validation of the noise emitted from a wind farm.

Compliance testing of an operational wind farm does not necessarily lead to a regression curve that follows the pre-construction regression curve, in that the derived curve tends to be linear or has a second order polynomial where the $x^2$ component is negative.

Figure 3 provides the synchronized background noise versus hub height wind speed measurements ("data") for the Cape Bridgewater wind farm [50] for house 88 (green scatter plot), in a relatively windy location, where the wind is the dominant noise source. The linear (red) and second order polynomial (blue line) regression lines of the data are then plotted for the region between the cut-in wind speed of the turbines (3 m/s), to a wind speed of 16 m/s. Due to the limitations of the data, the dotted red and blue lines represent the extrapolation of the linear and second order polynomial regression lines to a hub height wind speed of 23 m/s, respectively.

The 2003 version of the SA Guidelines [36], identifies that, where wind farm developers enter into an agreement with the owners of private land suitable for a wind farm site (where a level of compensation is made between the parties), the EPA cannot ignore noise impacts on the basis of an agreement.

The guideline identifies that an agreement is unlikely to be unreasonable if:

"the likely impact of exposure will not result in adverse health impacts (e.g., the level does not result in sleep disturbance)."

The EPA Guidelines identifies for a "host" that sleep disturbance is an adverse health impact. However, there is no criterion provided in the EPA Guidelines to qualify what noise levels constitute sleep disturbance, or what noise levels constitute adverse health impacts.

The guidelines indicate the need for protection from sleep disturbance for owners of private land suitable upon which wind farm may be developed but does not provide identification in terms of health impacts for residents who do not have a financial agreement with a wind farm developer.

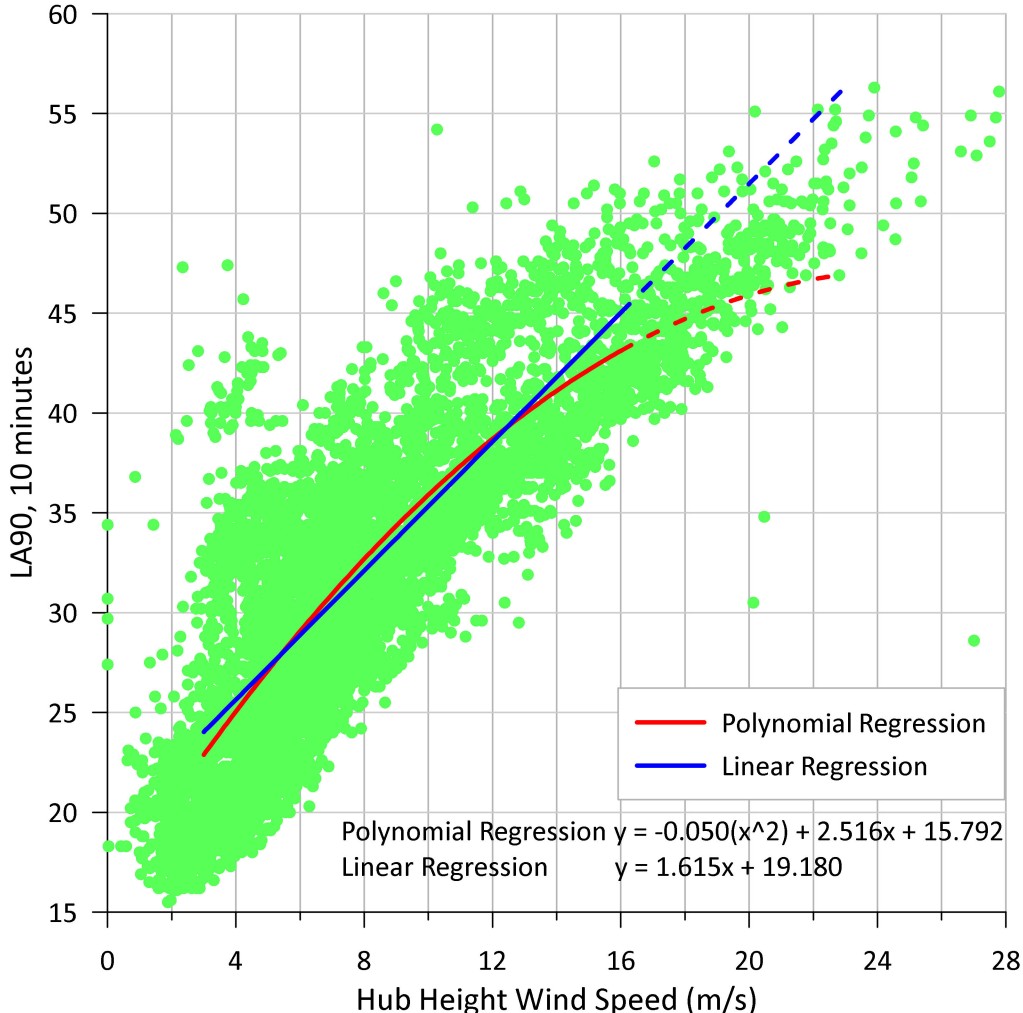

**Figure 3.** Regression Analysis of data for House 88 in Cape Bridgewater study [50] with permission from The Acoustic Group, 2015.

The SA EPA updated the guidelines in 2009 [37]. With respect to technical differences:

- The core objective did not change.
- The base limit of 35 dB(A) was allocated to receivers in localities primarily intended for rural living and 40 dB(A) for revivers in localities in other zones.
- The "host" adverse health impacts (e.g., sleep disturbance) were unchanged
- Background measurements for relevant receivers required for where predicted levels exceed the base noise level up to wind speeds to the rated power (previously up to 10 m/s)
- 2000 data points as previously, with the requirement for at least 500 points collected for the worst-case wind direction.
- Wind speed on the wind farm at hub height (previously at 10 m AGL on wind farm).

In 2019, the SA EPA issued draft (for consultation) wind farm environmental noise guidelines [38]. The basic parameters of the draft version are the same as the 2009 version of the guideline. Relevant to the topic of this advice is the preferred method for the calculation of wind farm noise, which is based on the logarithmic subtraction of the acquired background noise level (for the worst-case wind direction) from the combined noise level measurements, for each wind speed bin. The background level for the calculation of the wind farm noise should be derived from the downwind conditions.

Australian Standard 4959-2010 Acoustics—measurement, prediction, and assessment of noise from wind turbine generators [39] references ETSU-R-97. The Standard identifies the selection of noise criteria rests with the Relevant Regulatory Authority. The Standard presents (as examples) the use of the ETSU-R-97 regression line approach.

## 4. Fundamental Issues with Acoustic Compliance Testing

The purpose of acoustic compliance testing is to ascertain if the operational wind farm satisfies the planning (or design) limits. From the published guidelines, it would appear that failure to achieve the acoustic targets can result in adverse impacts to nearby communities.

Compliance testing in terms of absolute levels can be undertaken by on-off testing. Theoretically, one would be able to have a direct comparison of the ambient noise and the cumulative noise of the wind farm. However, shutting down an entire wind farm is not as simple as turning an off switch, as the sudden reduction of power into the grid presents issues, and in view of the costs and complexity in undertaking shutdowns, the wind farm operator prefers to have control of the process.

If one is to ascertain noise levels for integer (or specified) wind speeds, there may be difficulty in obtaining the required weather conditions (wind speed and receivers being downwind of the wind farm). To satisfy permit conditions, there may be a requirement for multiple shutdowns.

Compliance testing, with respect to the $L_{den}$ metric, presents issues due to the different weighting factors for the evening and night periods. The determination of the noise level as an on-off testing would require multiple tests throughout the monitoring period. If the compliance test is on the basis of determining the emission of a wind farm and then assuming the wind farm operates at that power all day, then the compliance becomes a theoretical process with assumptions that may not be valid.

Another issue that presents difficultly in compliance testing is the actual power output of the wind farm versus the rated power output (or specified wind speed), for which the acoustic design level applies. On-off compliance testing in Australia of the Capital wind farm in NSW [59] determined acoustic compliance for an average power output of 300 kW, for a wind farm with a rated power output of 140.7 MW. On one single test at the capital wind farm, at low wind speeds, the authority considered full acoustic compliance for all wind speeds.

The use of the regression analysis method for compliance testing overcomes the issue of shutting down a wind farm and seeks to undertake the monitoring of the background noise level at receivers versus wind speed at the wind farm, by comparing post-construction versus pre-construction background noise levels.

The fundamental concept, presented as a difference between the two regression curves, would indicate or permit the derivation of a noise contribution from the wind farm.

If one is seeking to assess the worst case scenario for residential receivers, there can be difficulty in undertaking compliance testing in a two week period to obtain sufficient data, having a wind direction spread of 45° either side of the direct line between the nearest turbine and the relevant receiver, or the concept of downwind measurement data proposed in the SA EPA 2019 draft guideline.

In some cases, rather than a two-week sample period, the compliance monitoring has tended to be over an extended period of time to obtain sufficient noise/wind data, primarily when there is a noise issue identified by residential receivers.

Neither the SA EPA Guideline or the New Zealand Standard identify the need for consideration of the different acoustic environment at different times of the year, where seasonal variation in the wind and weather patterns can alter the acoustic environment.

Examples have occurred of post-construction testing resulting in lower background levels (than for pre-construction), resulting in a reduced regression curve. It is unlikely that the building of a wind farm will reduce the ambient noise in an area, yet test results have been used exhibiting such results [59], with a claim of full acoustic compliance by comparing post and pre-construction regression curves.

It has been observed in Australia for some wind farms that compliance testing or background level monitoring for wind farms has been scheduled to occur in summer months, with attended measurements and observations identifying inaudibility of the wind farm.

However, in the rural landscape of Australia, it is not uncommon in the summer months to have elevated ambient noise levels as result of cicadas. This is similar to monitoring at night in rural or forest areas that reveals noise levels from insects and crickets to have an emphasis in the 4 kHz octave band, resulting in significantly elevated A-weighted background levels.

Neither versions of the SA EPA guideline or the New Zealand Standard look to a procedure for identification of the noise floor of the instruments used for the purpose of monitoring the evaluation of wind noise on the microphone. Unattended noise logging can (for class 1 m) have noise floors below 15 dB(A)—which is a significant improvement over the use of 1990s Class 2 instrumentation permitted by ETSU-R-97.

Both the SA EPA guideline and the New Zealand Standard suggest that monitoring at residential receivers use a microphone at a height of 1.5 m above the ground, but should exclude any relevant data where the wind speed (at the microphone) is greater than 5 m/s. This is an adaptation of general criteria for industrial noise or residential noise disturbance in Australia, where one is seeking to refer to an average minimum background level.

For the purpose of noise monitoring of wind farms in Australia, there has been a tendency to develop and utilize windscreens larger than that used for normal environmental noise assessments, and/or the provision of secondary windscreens, so as to reduce the effect of wind on the microphone.

For post-construction testing, there is a general requirement to utilize the same locations for noise and wind measurements as for the pre-construction testing.

However, there are examples of utilizing different wind heights on the wind farm between the two sets of results, and situations where there are residential receivers (the subject of complaints) that were not included in the pre-installation phase, thereby requiring data from another source (estimate) to be used.

The SA Guidelines/NZ Standard identify the need for photographic evidence of the installations, to identify proximity to trees or other objects that could influence the measurement results. However, there is no requirement to compare the photographic evidence between the different sets of acoustic measurements.

There are questions as to the ability of the regression method to ascertain the dB(A) levels of turbines when conducting measurements removed from the wind farm.

The use of the $L_{90}$ level removes the influence of the audible "swish" or "thumping" noise, as it is not a constant noise, thereby questioning the basis of allocating a 1.5–2dB adjustment to obtain an $L_{eq}$ level.

A fundamental issue in deriving the wind farm noise level by way of the regression analysis method is the influence of wind on the measured levels, as the wind affects the level of noise emitted from a wind farm and the ambient background level.

There have been studies with respect to measuring wind turbine infrasound and low frequency noise, in relation to the attenuation of wind directly on the microphone; various size windscreens are identified in the US ANSI/ASA Standard S12.9-2016/Part 7 [60]. Hansen and others have undertaken investigations of double layered wind screens [61–65], for low frequency and infrasound wind turbine measurements.

For general environmental acoustics in Australasia, where the acoustic assessment is based on an $L_{eq}$ level, compliance testing avoids winds above 5 m/s at the monitoring location—even using typical microphone windscreens of 50 mm–100 mm diameter. Such a limitation is not a suitable basis for testing wind farm noise, where it is essential to have wind for the turbines to produce electricity.

However, the focus for this technical article is not protecting the microphone from wind onto the microphone, but to consider the impact of the wind on the ambient background noise, where the

ambient background noise is the all-encompassing noise from near and far, but excluding the noise from the source under investigation (wind turbines), as required by BS 4142 [66].

Field observations during extended measurements for wind turbines have identified the presence of wind that can enhance or reduce ambient background levels where there is a distant noise source (e.g., a highway or surf/ocean). Monitoring in open paddocks near timbered areas reveals the presence of wind in trees (but not at the microphone), that creates a clearly audible noise that can influence or control the background level. Similarly, monitoring near dwellings that have bushes and trees nearby has revealed the presence of wind can influence the background levels.

For wind farm compliance testing in Australia (or the UK), where the regression analysis method is used, neither the SA Guideline or the New Zealand Standard consider the noise contribution from wind on the microphone or noise from trees in the derivation of the wind farm noise contribution.

As a result of seasonal variations throughout the year, one could expect different wind/weather patterns, that in turn would result in different ambient "base" background level curves.

The use of "wind farm" wind data from meteorological masts external to the wind farm (wake free data) does not represent the wind experienced across a wind farm and results in an incorrect representation of the sound power level being generated by a wind farm.

In considering the impact of wind on the measurement results for wind farms in Australia and the challenges in deriving the wind farm noise contribution, a series of case studies are presented.

## 5. Case Studies

Wind farms in Australia are generally located in rural areas, where there is an expectation of low ambient noise levels throughout the entire day. As a result of low ambient levels, the issue of instrumentation noise floor and the use of A-weighted measurements requires consideration of the instrumentation used.

ETSU-R-97 identified the instrumentation available at the time as being limited below 35 dB(A). In 1990, the specification of Class I precision sound level meters and statistical analyzers capable of determining the $L_{A\,90}$ level were often quoted as having a noise floor in the order of 26 to 30 dB(A).

Class II instrumentation utilized for logger measurements at that time had noise floors in the order of 30 to 35 dB(A).

Despite the limits of instrumentation in the mid-1990s to quantify the background level, it is an undeniable fact that remote rural properties can experience external ambient background noise levels at night less than 20 dB(A), which therefore would present an acoustic disturbance if there is a base criterion for the subject noise source in the order of 35 dB(A) or 40 dB(A).

Examination of Figure 2 would suggest that there is a noise floor for that measurement data in the order of 25 dB(A). The graph does not identify whether the noise floor relates to the acoustic environment of the area or the instrumentation used. The SA EPA Guideline (from which Figure 2 has been extracted) does not raise any issue in terms of noise floors.

In the 1990s, there were special microphones and preamplifiers used for low noise level assessments in laboratories that could go down to below 0 dB(A), but such laboratory microphones are not normally utilized in outdoor environments, because of the cost of such instrumentation, potential damage and the normal laboratory operating procedure to not use such microphones in the field.

The issue of noise floors for the instrumentation used to monitor ambient noise levels in rural environments (for an evaluation of potential windfarm applications) should be explored. As a result of reviewing proposed wind farm applications or assessing noise complaints of operational wind farms, a series of case studies have been undertaken, to examine the acoustic environment of the area in rural locations.

### 5.1. Case Study A-Proposed Hallett 3 Windfarm

The Hallett Wind farm is the collective name for four wind farms near the town of Hallett in South Australia (Coordinates 33°22′4″ S 138°43′43″ E). In 2012, there was a proposal to construct the Hallett

3 Windfarm [67], in the vicinity of Mount Bryan in South Australia. There was significant community opposition to the application and an Appeal to the Supreme Court of South Australia.

To ascertain the ambient background level with respect to the proposed wind farm, testing was undertaken at a rural residential property approximately 2 km south of the proposed Hallett 3 Windfarm. The location is well removed from any main roads that could influence the ambient background levels, that were found to be below 20 dB(A) during the daytime. Attended measurements at the site experienced background levels below 15 dB(A) in the evening and night-time. How much below 15 dB(A) could not be ascertained by the use of a SVAN 957 sound level meter, as the background level indicated on the meter is less than the specified electrical noise floor of the ACO Pacific microphone.

Observations at site indicated (in an aural sense) extremely low ambient background levels, but also that one could hear noise associated with wind passing through trees, where the noise was not that of the wind itself, but of the rustling of leaves in the trees.

The issue of the true ambient noise level for a remote rural area had been raised previously with respect to other windfarm applications.

### 5.1.1. Measurement Methodology

In addition to the standard two weeks of monitoring data as required by the SA EPA Guideline [37], the opportunity was available to conduct ambient monitoring in different locations in the remote area near the proposed Hallett 3 Windfarm, to investigate the change in ambient background level, depending upon the proximity of the noise logger microphone to trees.

The SVAN 957 sound level meter utilized an ACO Pacific microphone Type 7052, and for the purposes of outdoor measurements, was mounted on a stand where the microphone was oriented in a vertical direction and the diaphragm placed a height of 1.5 m above ground. Around the microphone was a perforated circular metal screen with a rain hood on the top of the circular screen. The screen has an air gap of 15 mm round the microphone preamplifier body and the microphone, with a GRAS AM0009 outdoor windscreen. Around the outside of the windscreen is a wire cage, to protect the microphone from bird damage.

Ambient measurements were provided in overall A-weighted levels and octave bands in 10 min increments. Wind measurements were undertaken at microphone height, using a Rainwise wind data logger time, synchronized using the same 10 min increments as for the sound level meter.

The basic noise logger microphone installation used today (by The Acoustic Group) is the same as that used for the Hallett investigations, but may incorporate SVAN 979 Sound level meters and GRAS 40AZ microphones to permit measurements down to 0.5 Hz.

In recent times, the original installation has been improved in terms of wind noise capability, by utilizing pantyhose stretched around the external metal cage.

### 5.1.2. Measurement Results

Preliminary measurements were undertaken, where the noise logger microphone was located 10 m from trees that varied in height from 3 to 7 m above the ground. The noise logger was in an open grassed area, where the trees were located 10 m to the eastern side of the microphone, i.e., having a 270° open field of view. The nearest tree to the west was in excess of 100 m.

The wind logger was located 5 m from the noise logger in an open area, as required for meteorological measurements. Due to topography and the presence of bushland to the east of the noise logger, the noise monitoring location was shielding from the prevailing wind.

Figure 4 presents the synchronized background noise level versus wind speed measurements ("data") for the measurement location 10 m from trees (blue scatter plot), with linear (black) and second order polynomial (red) regression lines of that measurement data. The results in Figure 4 indicate that the output of the noise logger was producing a noise floor in the order of 12 dB(A), which is below the microphone specification of 16 dB(A), but exhibits as a second order polynomial a curve that flattens

when the wind speed is less than 2 m/s at the microphone, to indicate the minimum ambient noise level at that position.

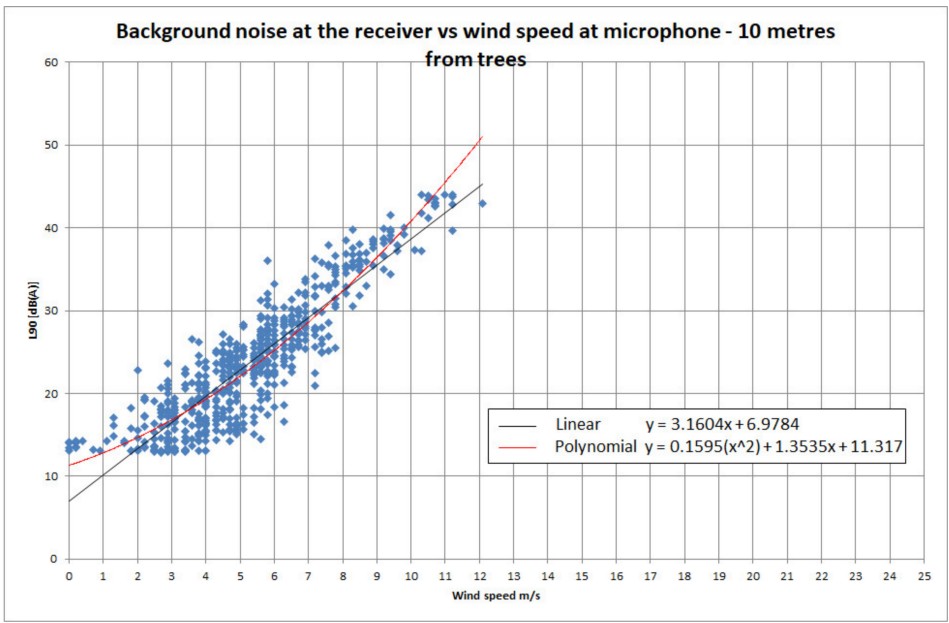

**Figure 4.** Synchronized background noise level versus wind speed at microphone for measurement location 10 metres from trees.

Figure 5 presents the synchronized background noise level versus wind speed measurements ("data") utilizing the same instrumentation with the logger being located in the middle of an exposed field, where there were no trees within 500 m, and the ground was furrowed in preparation for planting of a wheat crop (blue scatter plot). The logger was not installed whilst the ploughing operation was in progress. Figure 5 includes linear (black) and second order polynomial (red) regression lines of the measurement data.

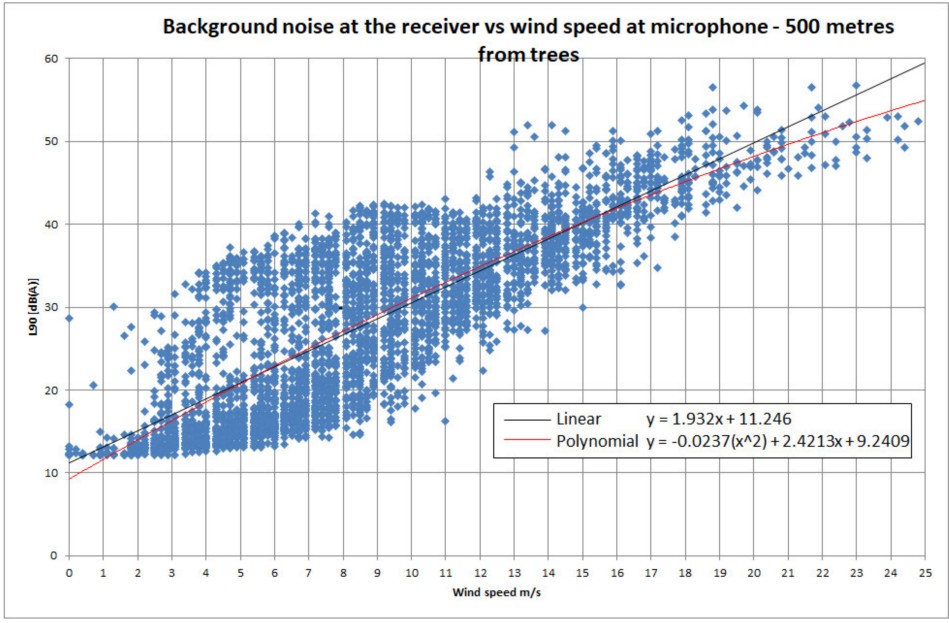

**Figure 5.** Synchronized background noise level versus wind speed at microphone for measurement location 500 metres from trees.

The results in Figure 5 cover a five-week period of monitoring and indicate a relatively close correlation to a linear result. It is noted that, on the exposed hillside, the logger was subject to higher wind levels than the other location.

An examination of the two graphs suggest that in proximity to trees, there is a flattening of the curve for the lower wind speeds that approaches the ambient background level, that could be attributed to the base ambient of the area or background noise from the animals/insects etc. habituating the treed area. The location at 500 m from trees indicates the wind noise of the system from 3 m/s, to say 15 m/s is a linear result.

In dealing with the occurrence of wind for the two locations, it is apparent that the ambient background noise at the location near trees is higher—as a result of the noise from the vegetation subject to the wind.

Figure 4 has the capability of determining the contribution of the monitoring system from the wind in proximity to the microphone, rather than the noticeably different slopes that occur where the noise logger is in proximity to trees, and is therefore receiving a noise contribution from the trees (as a result of the wind), that influences the measured background level.

The above material was presented to the 5th International Conference on Wind Turbine Noise [68], as a means of raising issues with respect to the accuracy of a regression analysis and the difficulty of identifying the actual wind turbine noise.

### 5.1.3. Discussion

In determining the measurement system's ability to determine the contribution from the wind, the regression analysis methods set out in Figure 4 suggests that, for a windspeed of 10 m/s, the regression line gives a level of 38–40 dB(A). However, if a general noise logger can, from Figure 4, have a wind noise contribution of 30 dB(A) at 10 m/s, then the regression line in Figure 4 at 10 m/s is identifying an ambient level, not from the wind, but from vegetation subject to the wind.

For wind turbines, one typically finds a requirement to exclude data at a receiver location when the wind at the microphone exceeds 5 m/s, on the basis of the wind contaminating the results. However, the determination of that criteria in a historical sense was developed a long while ago, with different instrumentation and no examination of the effect of wind on actual measurements.

Taking the material in Figure 5 and considering a limit 10 dB(A) above the linear line, then one is able to determine whether wind noise for this logger installation is affecting the results. In view of the manufacturer's specification for the sound level meter, it is considered appropriate to dismiss the design slope below 15 dB(A) and consider the data below 20 dB(A) as approximate. Establishing the level of wind noise below 3 m/s would need to be assessed using low level microphones, which is an expensive exercise.

Whilst in the strict sense, one is unable to logarithmically take an $L_{90}$ level from a $L_{90}$ level (in a time varying noise environment), it would appear that it is a general concept that has been adopted for wind farm compliance testing.

When one can identify the noise floor of the monitoring system versus wind levels conducted in an extremely quiet environment, then, from that material, one can validate the ambient noise levels at the pre-construction wind turbine phase, as a result of wind on the surrounding environment.

As a result of the above testing, it has been necessary to document/observe the presence of trees and bushes at various sites, for both the pre-installation and post installation of a wind farm.

### 5.2. Case Study B- Influence of Trees on Background Levels

Guidelines/Standards for wind farms in Australia suggest a residential noise monitoring location between 5 m and 30 m of a dwelling. However, there is no identification of the permitted proximity to trees/bushes.

In light of the experiments identified in Case Study A, for this technical advice, additional measurements were undertaken, to examine ambient noise from trees for a position of 1.5 m above the ground and 10 m above the ground, with wind measurements at the same corresponding heights. The addition of noise monitoring at 10 m above the ground is to examine the difference between the noise levels at the two different elevations, and also to permit an examination of the difference in the wind speed between 1.5 m position above ground and a 10 m position above ground.

The issue of the difference in wind speed is relevant in that, under meteorological requirements in Australia, the wind speed measurements are referenced to 10 m above ground level, and not the 1.5 m above ground level used for determining "wind impacts" upon residential noise monitoring locations in proximity to a wind farm.

In the absence of on-site wind farm wind speeds (either at 10 m AGL or hub height) there is difficulty in ascertaining acoustic compliance with the regression line curve that was determined at the pre-installation stage.

Where one is able to obtain the meteorological mast data or wind data from the anemometers on top of the turbine themselves, there is benefit in undertaking, at the same time, wind measurements at 10 m AGL at a receiver location. A comparison of the windfarm wind data and the receiver wind data at 10 m AGL can then provide a baseline for assessment of future testing where there may be significant non-compliance issues.

5.2.1. Measurement Methodology

Measurements conducted in an elevated open paddock in a semi-rural environment, 90 min drive from Sydney (Coordinates 33°34′11″ S 150°37′37″ E).

In a 2 acre open paddock, noise logger microphones were set up at a height of 1.5 m AGL and 10 m AGL, in the middle of the paddock, and duplicated at the edge of the paddock 10 m (horizontally) from trees immediately adjacent to a 4 acre natural bushland environment (see Figures 6 and 7). For the location adjacent to the trees, the canopy of the trees was generally in the order of 15 to 20 m AGL. To the east of the site is a strip of forest approximately 2 km wide and then open farmland for 12 km.

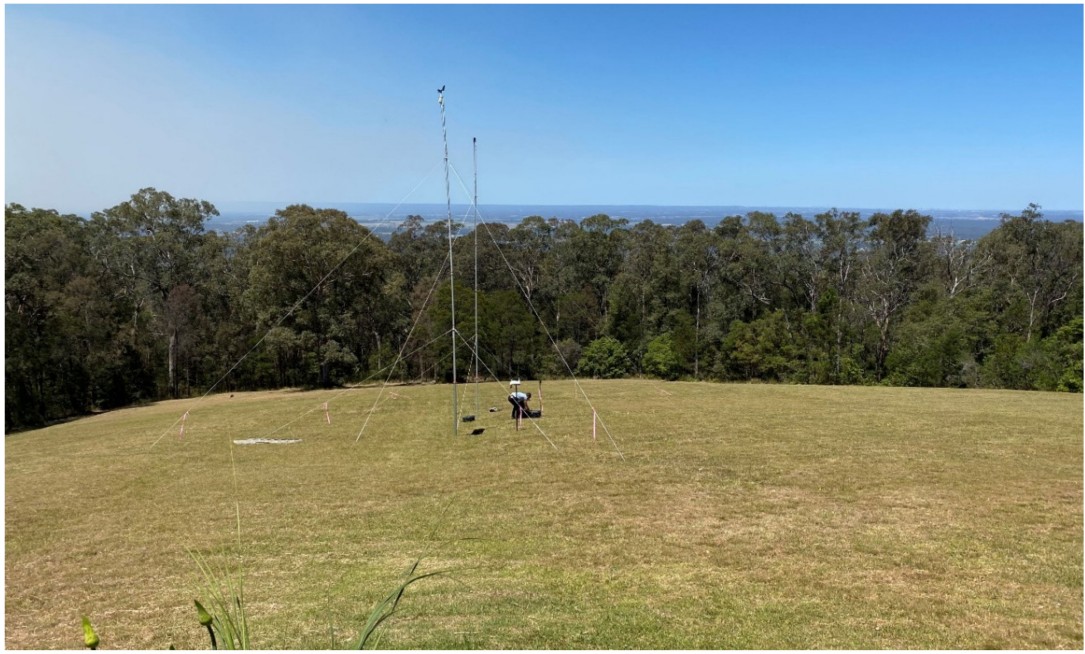

**Figure 6.** Center of open paddock.

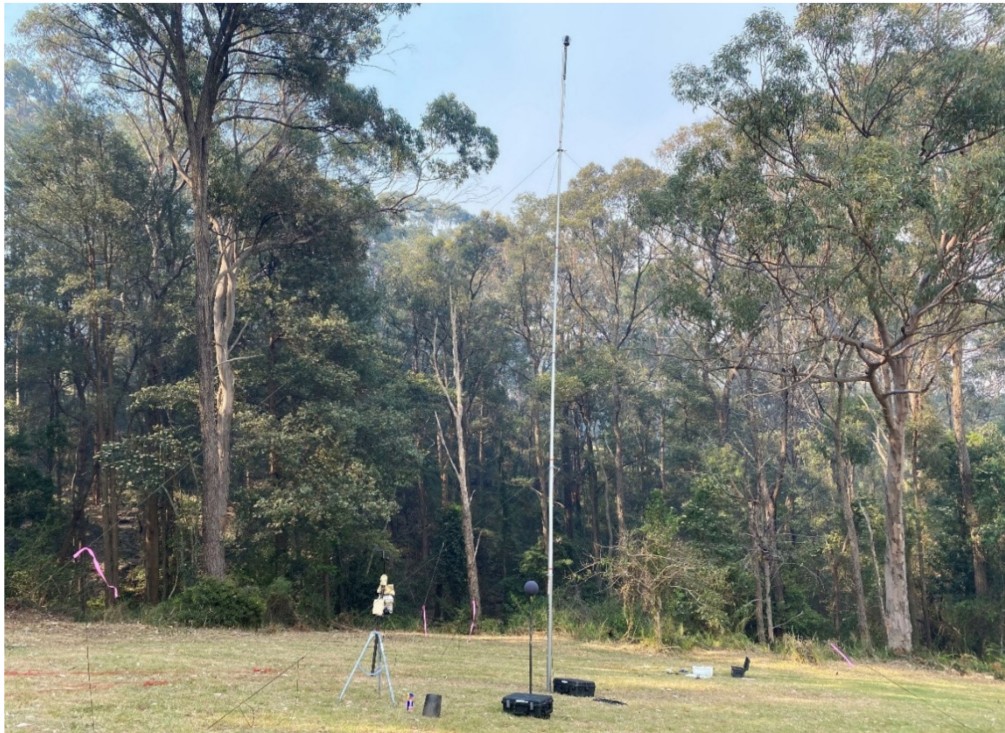

**Figure 7.** Noise Loggers at 10 from Trees.

The ambient sound level measurements were obtained using SVAN 957 sound level meters, with ACO Pacific microphones Type 7052 and wind screen arrangement as for Case Study A. The microphone was oriented in a vertical direction and the diaphragm placed a height of 1.5 m or 10 m above ground level.

Ambient measurements were provided in overall A-weighted levels and octave bands in 10 min increments. Wind measurements were undertaken at microphone height using Davis VUE weather stations, whilst at 10m above ground locations, the wind was recorded with Rainwise wind data loggers, time synchronized, using the same 10 min increments as for the sound level meter.

Testing was undertaken in the Australian summer period but was interrupted prior to Christmas (2019) due to bushfires in the locality, and then recommenced in the New Year.

Because of the severe drought conditions that existed prior to the bushfires, the acoustic environment during the summer period when the testing was undertaken was not subject to the typical increase in ambient noise levels as a result of cicadas.

The subject site is not influenced by wind turbine installations, but whilst being a relatively low acoustic environment, is subject to ambient noise from distant traffic in the valley below, and from an abundance of wildlife in the subject property and in the natural environment of an adjacent state forest.

5.2.2. Measurement Results

The noise/wind regression line components are of interest, to which the measurement data set out in Figures 8–11 provide a comparison of the results of 10 weeks of monitoring.

The green scatter plot in Figure 8 presents the synchronized background noise level versus wind speed measurements ("data") for the location in the middle of the paddock (10 m AGL), with a dark green line representing the third order polynomial regression line of that measurement data. The black line in Figure 8 represents the third order polynomial regression line of the $L_{90}$ for the individual wind bins (described in the next case study). The purple line below the data is the wind noise curve from the Case Study A (Hallett 3) testing, with the dotted blue line being 10 dB above the wind noise threshold, thereby identifying that the ambient noise, in this case, is not from the actual wind passing the microphone.

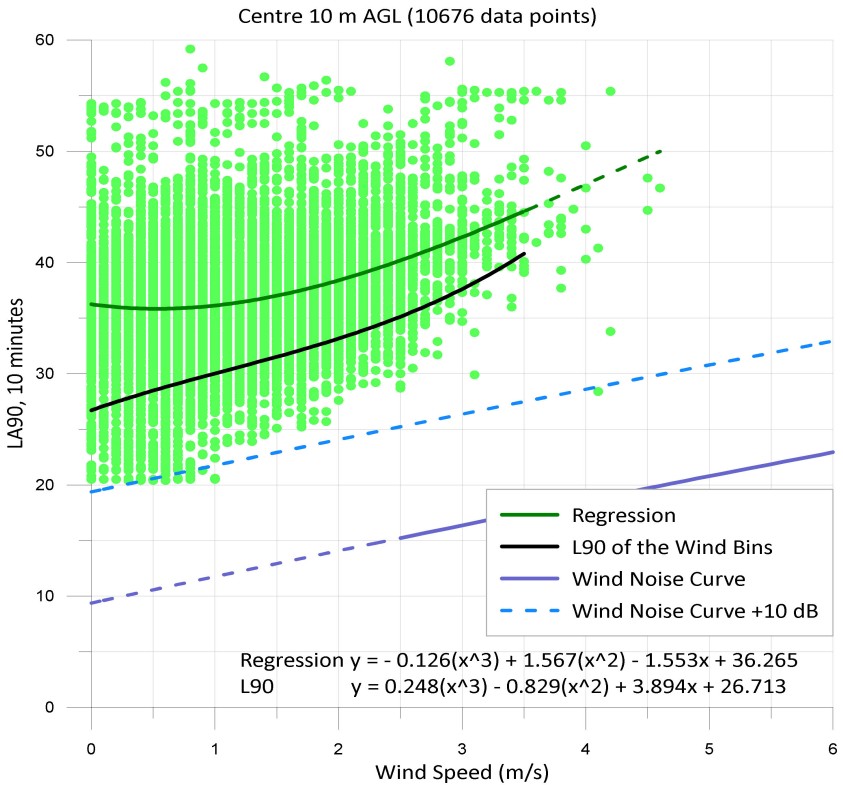

**Figure 8.** Ambient Regression Line—wind and noise at 10 m AGL, in center of field.

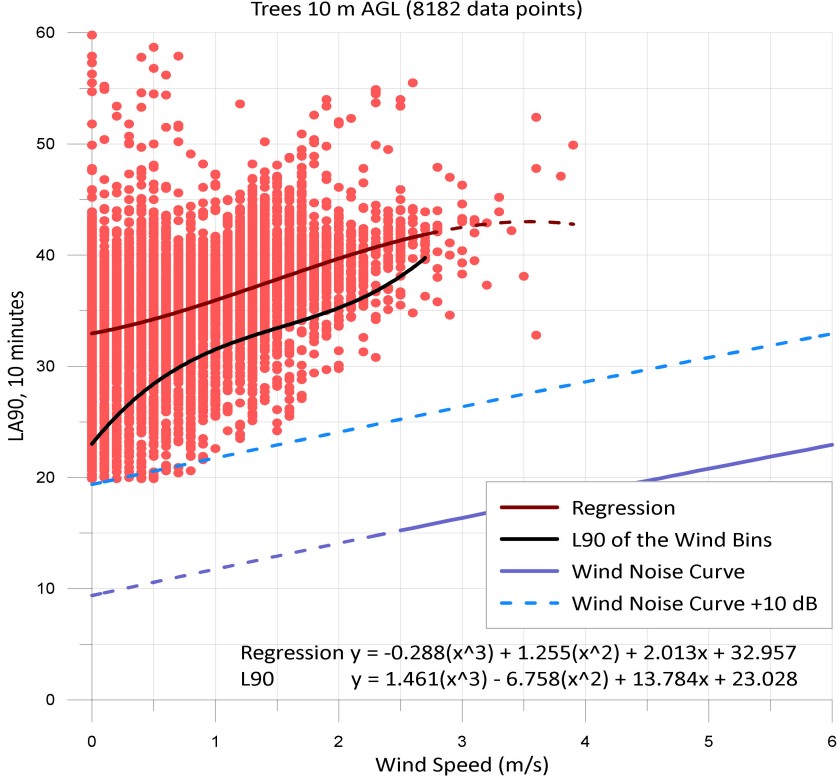

**Figure 9.** Ambient Regression Line—wind and noise at 10 m AGL, 10 from trees.

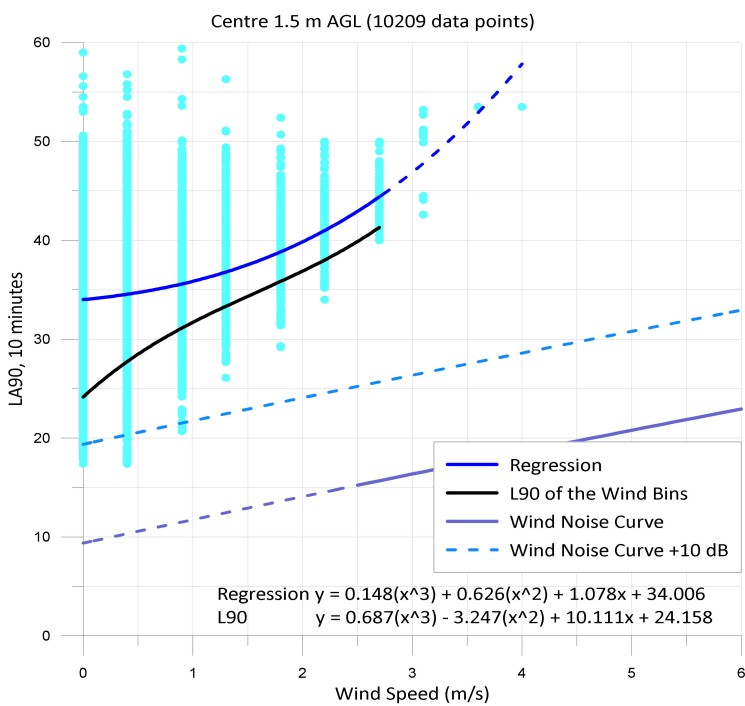

**Figure 10.** Ambient Regression Line—wind and noise at 1.5 m AGL, in center of field.

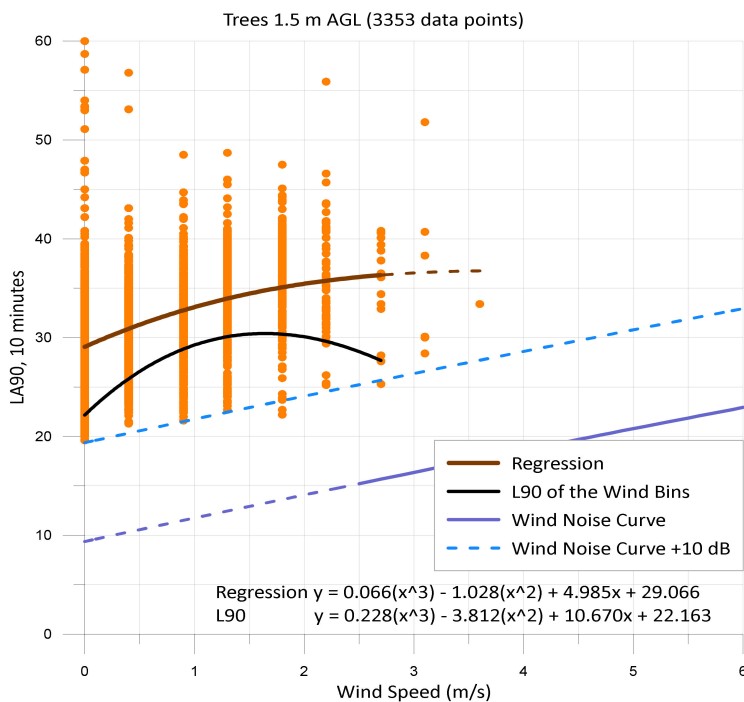

**Figure 11.** Ambient Regression Line—wind and noise at 1.5 m AGL10 from trees.

Similar plots are presented in Figures 9–11 for the measurement data for the different microphone locations. Data points in Figure 9 are for 10 m AGL at the edge of the paddock adjacent to trees (red). Data points in Figure 10 are for 1.5 m AGL in the middle of the paddock (blue). Data points in Figure 11 are for 1.5 m AGL at the edge of the paddock adjacent to trees (orange). The black lines in Figures 9–11 represents the third order polynomial regression line of the $L_{90}$ for the individual wind bins (described in the next case study).

5.2.3. Discussion

An examination of the regression line results reveals differences in the two sets of locations. The monitoring locations near the trees were subject to a lower level of wind, primarily as a result of shielding from the vegetation on one side of the open paddock.

The regression lines exhibit the typical flattening of a curve, as the wind levels are reduced, that reveals the repeatable minimum ambient background level.

The results in Figures 8 and 9 utilized a Rainwise wind data logger, whilst the results in Figures 10 and 11 utilized Davis Vantage Pro/Vue weather stations. The Davis weather stations are often use for residential receiver assessments. A comparison of the two figures reveal that the finer resolution of the wind direction of the Rainwise wind datalogger is preferred.

The results indicate that the elevated exposed location has higher minimum background levels, which is a result of a greater exposure to the surrounding area when compared to the corresponding ground-level locations.

The minimum night-time ambient background levels are influenced by noise from the forest areas, in proximity to the monitoring location.

Observations conducted during the multiple downloads of the data revealed for the locations in proximity to the trees that an increase in the wind gave rise to an increase in noise from the rustling of the trees.

Even in the center of the paddock, where there was a minimum distance of 100 m in all directions to the nearest trees, one could identify an increase in the ambient noise level when there were increases in the wind (even at low wind speeds). The increase in the noise level was attributed to the rustling of the leaves in the trees.

Whilst not provided in this document, the spectral differences between the location near the tree versus the center of the paddock are of interest, in that the regression lines when undertaken in octave bands exhibit different slopes. The difference in regression lines derived on a frequency basis is not apparent when undertaking measurements on a dB(A) basis and is the subject of further investigation.

*5.3. Case Study C-Overall Regression Analysis Versus Quadrant Analysis*

In Australia, for general environmental noise, the fundamental basis of assessment is to protect 90% of the people for 90% of the time. Both the Department of Planning in New South Wales [69] and the New South Wales EPA [70] have specified the use of noise logger data (excluding any data obtained where the wind speed of the microphone is greater than 5 m/s), to determine the individual daily L90 levels and then for a five or seven day period, to take the median of the individual days. This median is identified as the rating background level [69,70].

An intrusive noise target is then set at 5 dB(A) above the rating background level.

The concept set out in ETSU-R-97 (and referred to in Australasian wind farm guidelines) uses a polynomial regression line of wind speed versus the background level, to which there is then an application of a base criteria and/or background +5 dB, whichever is the greater.

For the ETSU-R-97 situation, whether one considers a linear or polynomial fit of the regression line, the curve represents approximately 50% of the population. If one then applies the regression background level plus 5 dB(A), is one only looking to protect 50% of the people 90% of the time? For the region where the base level of 35 dB(A) or 40 dB(A) is applied, the consequence of such criteria is protecting a much smaller percentage of the population for 90% of the time.

The NSW EPA identify that an intrusive noise target of rating background +5 dB(A) aims to result in the intrusive noise criterion being met for at least 90% of the time periods over which annoyance reactions can occur [70].

The use of the ETSU-R-97 regression line method for wind turbines in NSW fails to achieve the stated aim of the NSW Department of Planning and Infrastructure, where the general NSW objective is to set where possible noise goals are, to ensure that at least 90% of the population are protected from being highly annoyed for at least 90% of the time [69].

### 5.3.1. Measurement Methodology

Utilizing the same instrumentation used for Case Study A, the opportunity arose in 2012 to undertake three months of monitoring at a rural/residential property, in proximity to the Waubra Wind Farm (Coordinates 37°20′53″ S 143°36′1″ E), located in the southern state of Victoria, Australia.

The permit for this wind farm is expressed in terms of the 1998 version of the New Zealand Standard and specifies a background parameter as an $L_{95}$.

The SVAN 957 sound level meter utilized an ACO Pacific microphone Type 7052 and was mounted on a stand 10 m from the side of the dwelling, where the microphone was oriented in a vertical direction and the diaphragm was placed at a height of 1.5 m above ground. Around the microphone was a perforated circular metal screen with a rain hood on the top of the circular screen. The screen has an air gap of 15 mm round the microphone preamplifier body and the microphone, with a GRAS AM0009 outdoor windscreen. Around the outside of the windscreen is a wire cage to protect the microphone from bird damage.

Ambient measurements were provided in overall A-weighted levels and octave bands in 10 min increments. Wind measurements were undertaken at microphone height using a Rainwise wind data logger, time synchronized using the same 10 min increments as for the sound level meter.

### 5.3.2. Measurement Results

Figure 12 presents the synchronized background noise level versus wind speed measurements ("data") during the night-time period from three months of monitoring at a residential receiver in close proximity to the Waubra Wind Farm (blue scatter plot). In view of there being no access to the turbine data, the wind speed in this case is at the receiver location at a height of 1.5 m. The standard regression line process reveals the linear and polynomial fit (blue and red lines respectively in Figure 12) to be similar.

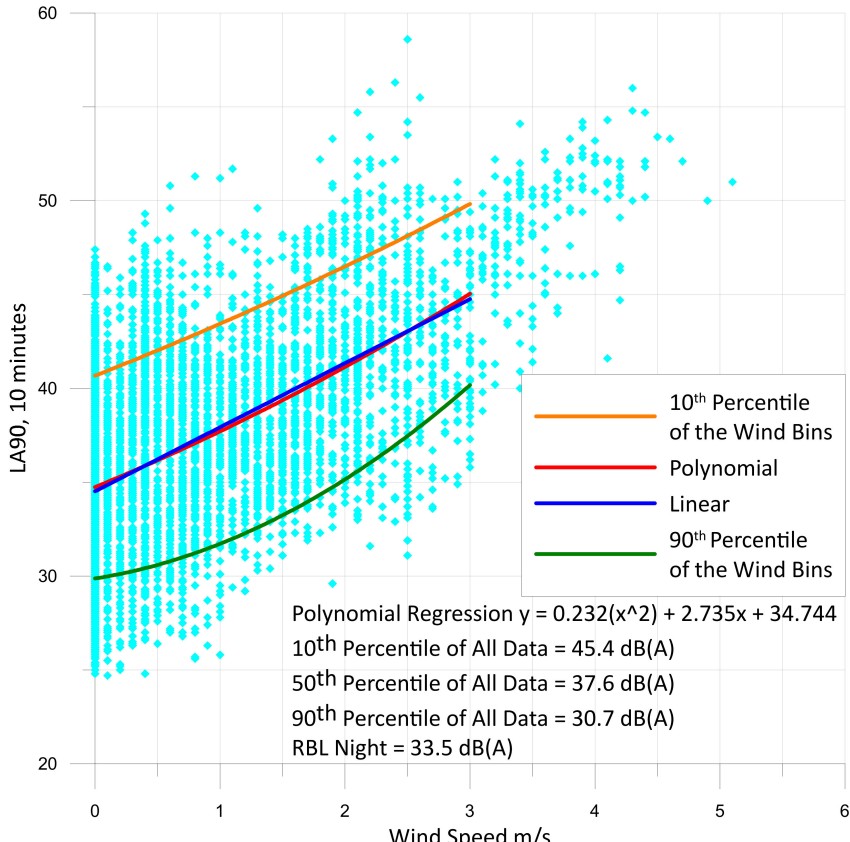

**Figure 12.** Waubra Wind Farm Nighttime Regression Curve-Post Construction, wind and noise at 1.5 m AGL, 12 June–11 September 2012.

The wind data (and corresponding time aligned noise levels) were separated into wind speed bins. A statistical analysis of each wind bin was undertaken to determine the 90th percentile of the $L_{A90, \, 10 \, min}$ level. Using the lowest 90th percentile in each wind bin permits the derivation of the true ambient background level across the wind speed (the green trace).

Figure 12 reveals a significant difference between the "regression background level" versus the L90 percentile line. Using the ETSU-R-97 concept, the regression line background +5 dB limit is more in line with the true Rating Background Level +10 dB(A).

If the pre-installation regression line is based on a conglomeration of wind speed and wind directions, then should a worst-case assessment be based upon all the data or assessing the results against a pre-installation regression curve for the critical wind quadrant?

The night-time wind speed and direction for the monitoring period are set out in Figure 13 for an analysis of all the individual wind direction bins, to determine the four principal quadrants based upon the dominant NNW nighttime wind in the monitoring period (winter).

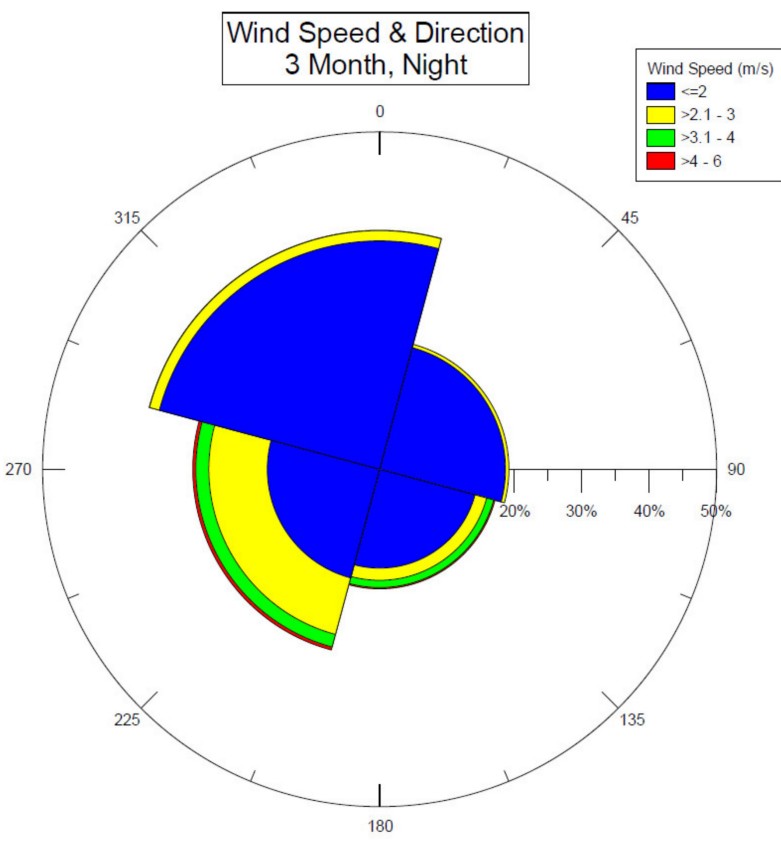

**Figure 13.** Waubra Wind Farm Nighttime Wind-Post Construction re Microphone Height measured wind data for wind direction quadrants.

The Guidelines/Standards in Australia recommend that the wind farm post-compliance test is to obtain at least 500 data points as a worst-case scenario for a wind direction, from the windfarm to the receiver. Restricting the regression method to 500 points does not (for the three months of data) have sufficient points to provide nighttime regression analysis (i.e., the monitoring period needed to be longer than 3 months) for the assessment of nighttime criteria.

Utilizing the subset of quadrant wind data and corresponding 24 h noise data allows the derivation of regression curves for the four principal wind quadrants, as shown in Figures 14–17.

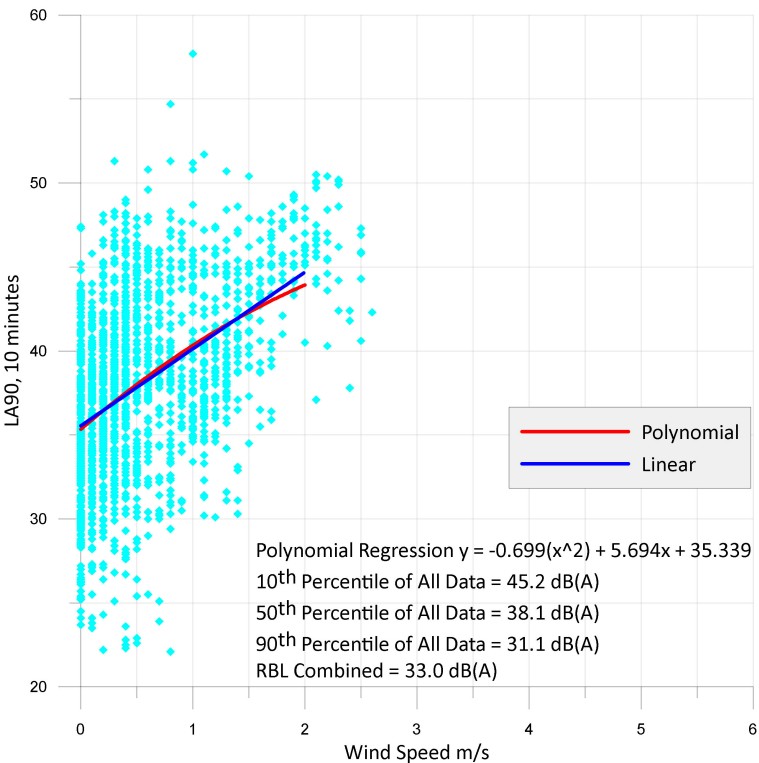

**Figure 14.** Waubra Wind Farm Regression for NE Wind Quadrant—Post Construction Test, Microphone and wind at 1.5 m AGL.

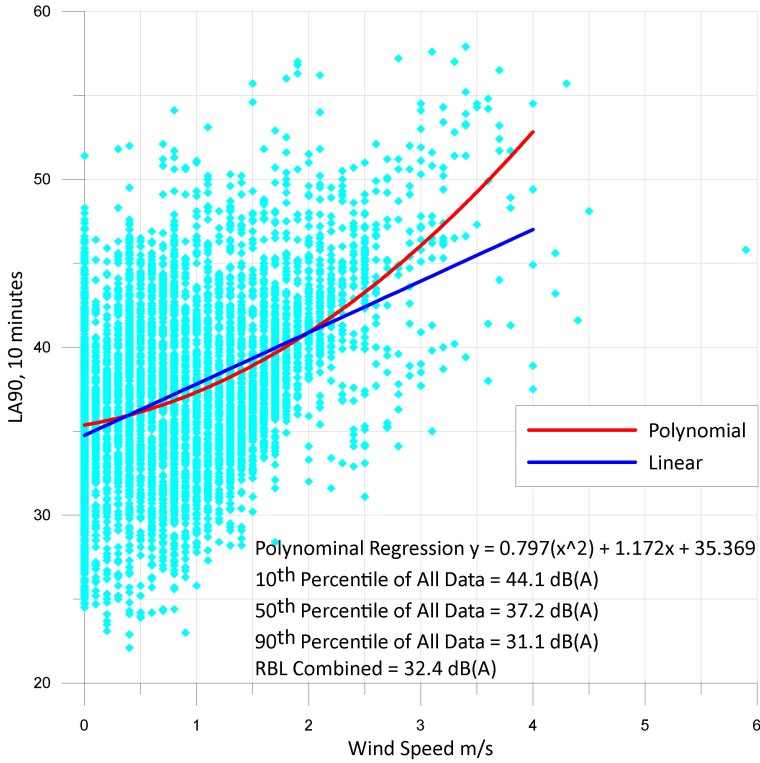

**Figure 15.** Waubra Wind Farm Regression for NW Wind Quadrant—Post Construction Test, Microphone and wind at 1.5 m AGL.

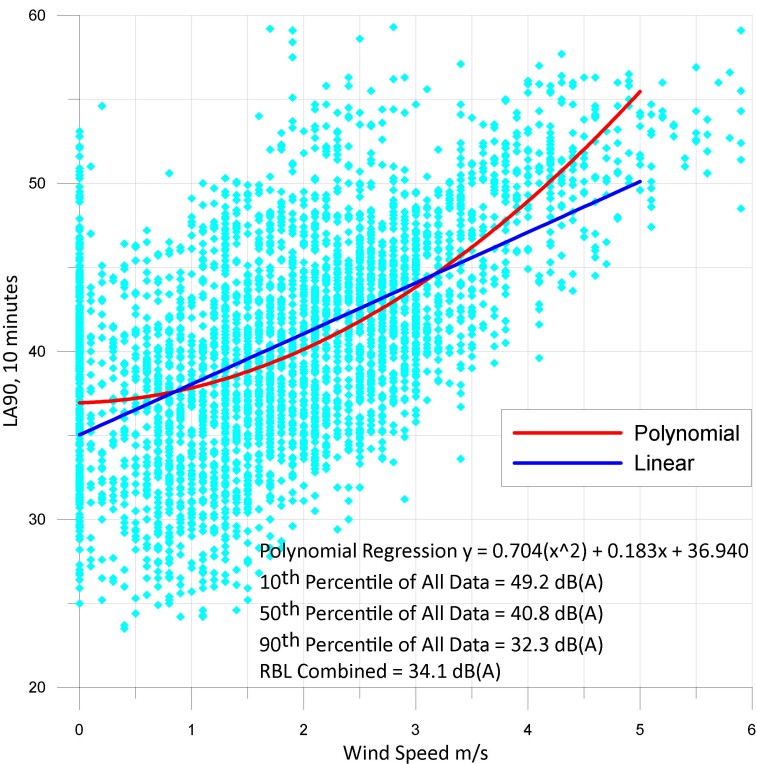

**Figure 16.** Waubra Wind Farm Regression for SW Wind Quadrant—Post Construction Test, Microphone and wind at 1.5 m AGL.

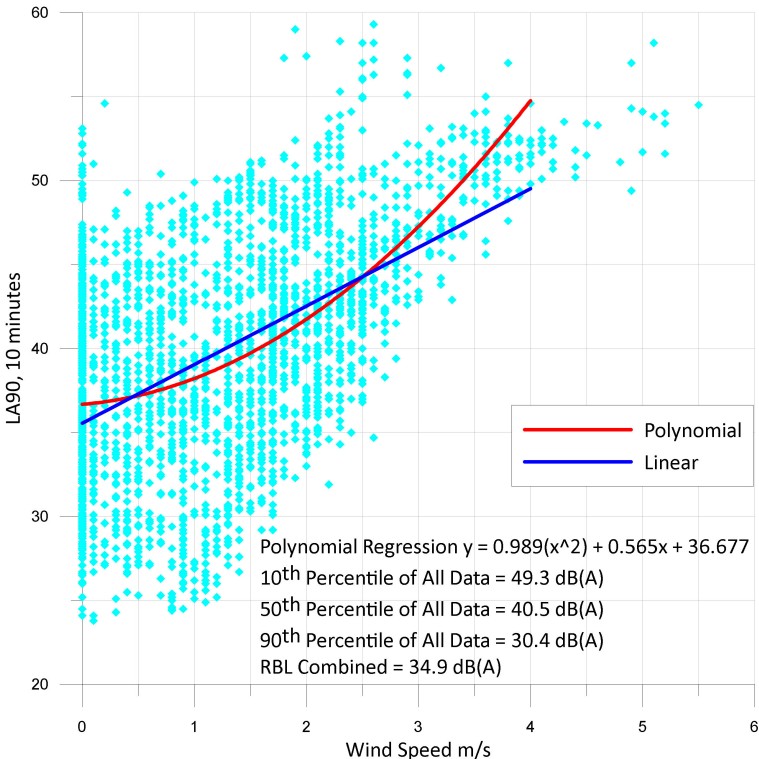

**Figure 17.** Waubra Wind Farm Regression for SE Wind Quadrant—Post Construction Test, Microphone and wind at 1.5m AGL.

### 5.3.3. Discussion

If compliance noise monitoring is restricted to only 2000 data sets, then does one have sufficient information for the purposes of evaluating a worst-case scenario?

If a pre-construction noise assessment for a windfarm only utilizes two weeks of noise data, then there would appear to be a limitation in terms of the statistical validity of the pre-installation regression line for the subject site. At the assessment stage, to determine the underlying background level, one must consider the different weather effects that may occur for different seasons throughout the year.

If there are different regression lines applicable to different season or different wind conditions, then the matter of determining acoustic compliance utilizing a base curve derived from two weeks of data cannot be justified as "appropriate criteria" for the entire operating life of a wind farm.

### 5.4. Case Study D–Wake Effects

At the planning stage, the prediction of noise levels in the vicinity of a proposed wind farm utilizes manufacturer's sound power level data for wind turbines, with Australian Standard AS 4519-2010 [39] nominating the level at the receiver location $L_R$ (at a height of 1.2–1.5 m) is:

$$L_R = L_w - 10 \log (2 \pi R^2) - \Delta L_a \tag{1}$$

where R = distance between the source and the receiver, in meters, and

$$\Delta La = \alpha aR$$

$\alpha_a$ = attenuation of noise due to air absorption, in dB/m.

The above equation relates to a single turbine and assumes free airflow to the turbine—being the basis of the testing to determine the sound power level.

The above equation does not require any allowance for any directivity pattern of the wind turbine.

The radiation pattern of a wind turbine is complex and has been the subject of various studies. Hubbard [29] and Okada [71] identified that the radiation pattern of a turbine is a broad dipole. Oerlemans [72] and Doolan [73] consider the trailing edge noise directivity of the turbine blade to be a cardioid pattern, but the leading-edge interaction and the blade-tower interaction to be a dipole pattern. On the basis of the above studies, there is, in a plan view, a directivity pattern. Such a pattern can give rise to an enhancement of the actual noise contribution from a turbine—when compared with the general approach of considering a point source or each side of the turbine radiating off the rotor diameter as a hemispherical source.

However, whilst the airflow before a turbine (inflow) may be undisturbed, after the turbine, the airflow (the wake) is disturbed with the disturbed air continuing downstream of the turbine. The interaction of the wake with downstream turbines results in a loss of power generated by the downstream turbines but can also increase the level of noise [74].

Dickinson [75] provided examples of spiral waves behind a propeller and a wind turbine blade. This wave occurs immediately downstream of the blades in which the effect of the rotating blades is dominant. Sedaghatizadeh [74] identifies the near-wake region can extend out to a distance of 5 times the rotor diameter. The far-wake commences where the flow is characterized by significant velocity deficit and increased turbulence. Barlas [76] identifies that the acoustic energy is redistributed downwind from a wind turbine, due to the fact that, in wake-induced flow field and for long wakes in stable atmospheric conditions, there can be considerably high sound pressure level amplification in the far field.

With the wake of a turbine extending well past a turbine [77], the interaction on other turbines as a narrow beam can significantly alter the sound output of downstream turbines for small variations in the wind direction and wind speed.

The examination of acoustic compliance reports prepared in Australia under the SA EPA Guideline or the New Zealand Standard finds regression curves to suggest acoustic compliance, but the compliance reports do not identify the wind parameters or the power output of the windfarm, or the corresponding wind speed and direction at 10 m above ground at the receiver location.

As the interference of wake effects from turbines on the wind flow across the wind farm is not modelled in the predicted levels for a wind farm, in the presence of wake effects, those turbines would give rise to different sound power levels and potentially significant differences in the propagation of noise from individual turbines, when compared with the predicted levels that area based on sound power measurements of a single turbine.

In considering the issue of wake effects and the non-linear operation of turbines on a wind farm with respect to acoustic compliance, the operational wind farm data are not placed in the public domain. Obtaining data in relation to the actual operation of the turbines on a wind farm to explain the different wind levels and turbine power outputs is difficult, and restricts the ability to relate physical observations at wind farms of different turbine directions and different rotor speeds, or validate the claims of acoustic compliance.

Obtaining the SCADA (supervisory control and data acquisition) for a wind farm provides the opportunity to examine the differences discussed above.

### 5.4.1. Cape Bridgewater Wind Farm

The report on the Cape Bridgewater Wind Farm [50] (located at Coordinates 38°22′35″ S 141°22′20″ E) placed examples of different turbine power output, blade angle, wind speed and wind direction in the public domain, to assist in understanding the variations in individual turbines across the site.

Identification of the averaged nacelle wind anemometer data and cumulative power output for all the turbines at the Cape Bridgewater Wind Farm was provided in the study., Additional examples of differences in wind and turbine output over time can be seen in the Cape Bridgewater Wind Farm Study (pages 80–86 of reference [50]).

Figure 18 presents the variation in wind direction for the southern portion of the Cape Bridgewater Wind Farm for one 10 min period. Figure 18 reveals that wind direction is not consistent across the wind farm. The different turbine directions indicate that the wakes would have different directions and suggests that the assumption of simply adding the number of turbines together (without the consideration of the results of the interaction of other turbines) may be incorrect.

Figure 18 identifies a meteorology mast (PMM29) that is on the southern boundary of the wind farm. From Figure 18, for some wind directions, the meteorology mast would be subject to wind affected by wake turbulence, whilst for a wind from the south, there would be no turbulence.

The wind data obtained at the pre-installation phase of the Cape Bridgewater wind farm project were from meteorological masts now within the wind farm. For the averaged wind data of the turbines in Figure 18, the corresponding wind speed from the meteorology mast was higher than the averaged wind across the southern section of the wind farm.

### 5.4.2. Discussion

Acoustic compliance test reports for wind farms in Australia refer to "wake free" wind data. For some wind farms, there will be multiple meteorology masts outside the wind farm, where, for individual groups of wind directions, wind data are excluded, and the remaining data become the derived "wake free" wind speeds to be used for acoustic compliance certification.

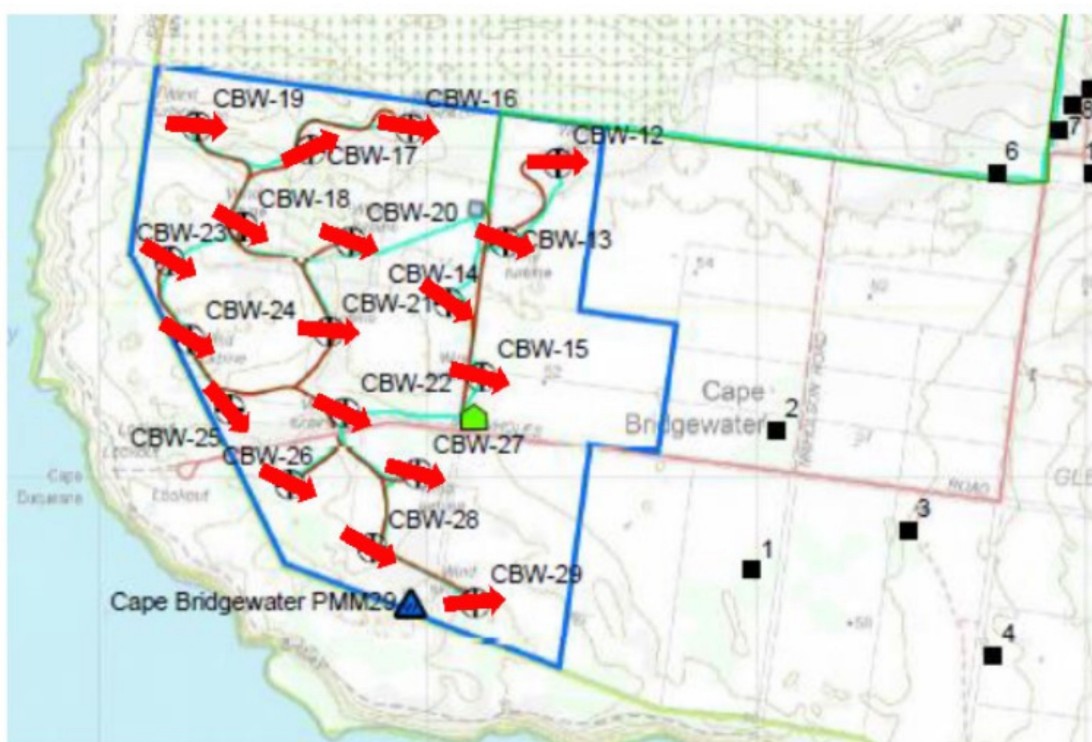

**Figure 18.** Cape Bridgewater Wind Farm (Southern end), Average wind direction 9.00 a.m.–9.10 a.m. on 10th June 2014.

The basis of how the "wake free" hub height wind dataset was derived is not normally provided, but on examining the SCADA (where it is available), the "wake free" data (in the absence of material to support the adjustments) becomes difficult to accept.

Erecting new meteorology masts around the external perimeter of the wind farm (say the four corners), and only using the unobstructed quadrants of wind data to obtain a combined "wake free" dataset, does not provide the same results as the average of the actual wind speed experienced by all turbines.

Examination of the wind speed across a wind farm (from individual turbine anemometers on top of the nacelle) reveals significant reductions in wind speeds. From the SCADA information, the average of the wind speed across the wind farm is lower than the "wake free" results. The use of the "wake free" wind data results in a shifting of the data regression curve to the right and makes compliance easier (for the wind farm) but does not represent the wind speed that the turbines experience. For the example in Figure 18, the meteorological tower wind speed was 1.9 m/s higher than the average nacelle wind speed for the turbines shown.

The regression curve method does not identify the direction of the wind with respect to the receiver location. This may conveniently remove the possibility of evaluating a downwind situation for a residential receiver, and thereby provide an inaccurate compliance test result that underestimates the wind farm noise level.

The first consequence of not taking into account the wake effect on down wind turbines leads to an underestimate of the potential power output of the wind farm, resulting in predicted sound levels (on the basis of reduced wind across the wind farm) that are too high.

The second consequence of not considering the wake effect on downwind turbines is the potential for some locations at time to experience noise levels higher than predicted (at nominated integer wind speeds), by reason of the focusing of the wake.

The third consequence of not considering the wake effect on downwind turbines is the potential for some locations to experience enhanced special audible characteristics (not identified in the planning stage), by reason of the significant turbulence on the inflow side of some turbines, narrow directivity patterns and topography effects related to the propagation of the sound (identified by Kelley in relation to the Mod-1 wind turbine).

In the derivation of the resultant regression analyses for the purpose of compliance, another factor that arises is an assumption that the windfarm is not operating in any restricted capacity and/or is operating in accordance with the operations specified in the development application that was assessed using the assumptions for the predicted levels. The predicted levels are normally evaluated under an assumption that the noise generated by the wind turbines are directly related to the sound power level provided by the manufacturer for the various integer wind speeds.

If there are restrictions in terms of the operating parameters for the windfarm whilst compliance testing is occurring, then there is the possibility that the regression curve determined as a result of that compliance testing is not the same as if the windfarm was operating in unrestricted capacity (i.e., being different to that nominated at the development application stage).

For example, if the windfarm is restricted to a low output (during the compliance test), then whilst the regression line is based upon noise levels versus windspeed at the wind farm (10 m AGL or hub height), that does not represent the "normal" operation of the turbines for that wind speed. When the turbines are not operating at their expected capacity and a regression line is determined for the 10 m AGL or hub height actual wind speed, the resultant regression lines are misleading.

Under the regression analysis method, the noise and wind data are averaged over a period of time (at least 2 weeks). However, the averaging covers a variation in wind that must, by definition, also give rise to a variation in the sound power level, and in turn, the electrical power output of the wind farm. If the wind farm for the compliance testing only produces on average 20% of the rated capacity of the wind farm, then what is the regression line representing? Do the compliance test results represent a noise curve for 20% capacity? The mathematics of the sound power level at reduced wind speeds identifies a lower sound power level, and therefore lower noise levels at residential receiver locations.

How does one obtain, from a 20% capacity result, that the wind farm is compliant under the worst-case scenario for full rated capacity? Should not the compliance results be scaled up by the same increment in the power levels for higher wind speeds?

Whilst the above discussion has focused on the regression method for determining acoustic compliance, the majority of the above discussion in relation to determining the actual noise contribution of the subject wind farm would also apply to the situation where there is an absolute noise limit or an absolute noise limit for a specified wind speed.

## 6. Conclusions

Noise guidelines/standards/policies for wind farms around the world have a range of noise levels, with the majority of the limits based on a version of an $L_{eq}$ metric using the A-weighted level.

In Australia, the assessment and compliance testing of wind farms is based on the regression line method originally developed in the UK.

Some wind farms in Australia and New Zealand give raise to noise complaints, to the extent that residents abandon their homes—despite the wind farm being "acoustically compliant" with the relevant permit conditions (based upon the regression line method).

Australian courts rely upon the guidelines produced by planning or environmental authorities—notwithstanding that there is no evidence based material to support the noise criteria specified for wind farms or verify that there will be no impact.

The operation of a wind farm requires wind. It is acknowledged that the presence of wind causes the ambient noise to increase.

The generation of inaudible wind turbine noise (i.e., contributions below the threshold of hearing) that gave rise to people (with an exposure to turbine noise over a number of years) being able to identify the operation of the test signal [78,79] has led to questions raised in this technical advice.

The Cape Bridgewater Study [50] identified a correlation of the A-weighted level with the wind speed, not the power output of the wind farm. It is relatively easy to identify unique narrow band spectral components associated with the operation of a wind farm. However, determining the A-weighted contribution of the wind farm to validate predicted levels is difficult.

Derivation of the wind farm noise dB(A) contribution requires separating the wind turbine noise from the wind affected ambient noise and the residual ambient noise [68,80].

The focus of this technical article has been to identify challenges to the derivation of the wind turbine contribution. From the perspective of psychoacoustics, it is essential to determine the actual noise contribution of a wind farm in the environment in which it occurs.

If one is unable to determine the contribution of the wind turbine noise as part of the existing ambient noise, then one is unable to evaluate the effectiveness of predicted noise levels.

If the predicted levels are incorrect, then the use of the A-weighted dose response curves provided by Pedersen et al. [42], Keith et al. [47], Davy [12], Janssen [43], Kuwano [44] or the WHO [9] may be inappropriate.

Similarly, sleep studies into noise from wind turbines could be determining the wrong threshold levels.

Two case studies have provided the results of testing to identify the A-weighted level of wind for the instrumentation used for the unattended logging of the ambient noise, and the identification of an additional wind based component of the ambient being related to wind on vegetation. The two case studies were free of wind turbine noise and reveal, that in low ambient environments, the influence of wind induced vegetation noise requires larger separation distances than suggested by Hansen [81] and Bolin [82].

Two case studies have identified issues with respect to compliance testing and question the ability to suggest a wind farm will be acoustically compliant at all times.

Issues with respect to the acoustic compliance testing of operational wind farms in Australia have identified that additional research is required in determining the actual wind farm noise contribution over a range of operating parameters before one can establish full acoustic compliance.

If one is unable to determine the noise contribution of a wind farm (inside or outside a dwelling) then for wind turbine sleep studies undertaken in a laboratory, what noise levels should be used?

The consequence of the determination of the real noise contribution of wind farms at receiver locations could require a review of noise levels that have been used (and are currently being used) for planning purposes and in psychoacoustic research into wind farm noise.

**Author Contributions:** S.C. and C.C. contributed 95%, 5% to conceptualization; 90%, 10% to field work; 50%, 50% to formal analysis; 95%, 5% to original draft preparation; 80%, 20% to review and editing and 50%, 50% to literature review. All authors have read and agreed to the published version of the manuscript.

**Funding:** This research received no external funding.

**Acknowledgments:** We acknowledge the technical support from Olivia Walker (undertaking an internship with TAG) who jumped into the deep end of the project, specifically in relation to the additional ambient noise logging, analysis and regression line assessment undertaken for this technical note.

**Conflicts of Interest:** "The authors declare no conflict of interest." The principal author has undertaken assessments of wind farm noise for residents and is the author of the Cape Bridgewater Wind Farm Study.

**Abbreviations**

The following abbreviations are used in this manuscript:

| | |
|---|---|
| AGL | Above ground level |
| dB(A) | Cumulative sound pressure level after passing through the A-weighting filter that approximates the response of the human hearing |
| ETSU | For the UK guideline ETSU-R-97 |
| $L_{A10}$ | A-weighted sound pressure level exceeded for 10% of the time |
| $L_{A90}$ | A-weighted sound pressure level exceeded for 90% of the time |
| $L_{A95}$ | A-weighted sound pressure level exceeded for 95% of the time |
| $L_{A90, 10\ min}$ | A-weighted sound pressure level exceeded for 90% of the time in a 10 min sample (840 s) |
| $L_{eq}$ | Energy averaged sound pressure level |
| $L_{den}$ | The day/evening/night-time level being a time weighted $L_{eq}$ incorporating adjustments for the evening and night-time periods as defined in Section 3.6.4 of ISO 1996-1:2016 |
| $L_{dn}$ | Day-night-weighted sound pressure levels incorporating an adjustment to the nighttime measured level, as defined in Section 3.6.4 of IOS 1996:2016 |
| $L_{night}$ | Equivalent continuous sound pressure level when the reference time interval is the night. |
| $L_{pA, LF}$ | A-weighted level between the frequencies of 10 Hz and 160 Hz |
| SERI | Solar Energy Research Institute |
| SA | The state of South Australia in Australia |
| NASA | The National Aeronautics and Space Administration |
| NSW | The state of New South Wales in Australia |
| SCADA | Supervisory Control and Data Acquisition |
| EAM | Enhanced amplitude modulation |

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
