# Peer review of "Determination of Acoustic Compliance of Wind Farms"

_acoustics, doi:10.3390/acoustics2020024_

Round 1
Reviewer 1 Report
The paper presents results of investigations of wind turbine and background noise that show limitations of currently used acoustic criteria for wind farm. The topic is important and the results are interesting but the paper has many unexplained statements and does not cover modern state of art in several discussed areas. I recommend the following improvements of the paper:
1.The paper introduction starts from the sentence about requirement to “take account of community reaction or studies into different noise sources” ( line 26) but only very limited information about investigations of wind farm noise to health effect is presented in the end of the paper (lines 701-705). There are a lot of publications on this matter and review of state of art in the area can be included to the paper. Few publications on this matter:
- Bakker, R.H., Pedersen, E., van den Berg, G.P., Stewart, R.E., Lok, W. and Bouma, J., 2012. Impact of wind turbine sound on annoyance, self-reported sleep disturbance and psychological distress. Science of the Total Environment, 425, pp.42-51.
- Abbasi, M., Monazzam, M.R., Akbarzadeh, A., Zakerian, S.A. and Ebrahimi, M.H., 2015. Impact of wind turbine sound on general health, sleep disturbance and annoyance of workers: a pilot-study in Manjil wind farm, Iran. Journal of Environmental Health Science and Engineering, 13(1), p.71
- Schmidt, J.H. and Klokker, M., 2014. Health effects related to wind turbine noise exposure: a systematic review. PloS one, 9(12).
- Onakpoya, I.J., O'Sullivan, J., Thompson, M.J. and Heneghan, C.J., 2015. The effect of wind turbine noise on sleep and quality of life: A systematic review and meta-analysis of observational studies. Environment international, 82, pp.1-9.
- Jeffery, R.D. and Brett Horner BA, C.M.A., 2014. Industrial wind turbines and adverse health effects. Canadian journal of rural medicine, 19(1), p.21.
2. The paper expresses doubts in the applicability of the ETSU-R-97 criteria being applied to current wind farm turbines having a capacity greater than 5 MW (lines 15-18). I recommend to include a description of reasons for these doubts. The applied acoustic criteria do not depend on the wind turbine power from physical acoustic point. I recommend to the authors to include brief description of physical phenomena of wind turbine noise and to try explain why the ETSU-R-97 criteria cannot be used for powerful wind turbines. I like NASA publication of this matter: Hubbard, H.H. and Shepherd, K.P., 2009. Wind turbine acoustics. https://ntrs.nasa.gov/archive/nasa/casi.ntrs.nasa.gov/20090026531.pdf
3. The main paper result is connected with assessment of influence of noise background and noise on microphones. There are many publications of this matter and state of art in this area can be included to the paper. The results of the paper are needed to be compared with the previous published data.
Few papers on the wind induced microphone noise:
- Van den Berg, G.P., 2006. Wind-induced noise in a screened microphone. The Journal of the Acoustical Society of America, 119(2), pp.824-833.
- Jackson, I.R., Kendrick, P., Cox, T.J., Fazenda, B.M. and Li, F.F., 2014. Perception and automatic detection of wind-induced microphone noise. The Journal of the Acoustical Society of America, 136(3), pp.1176-1186.
- Kendrick, P., Cox, T., Li, F., Fazenda, B. and Jackson, I., 2013, July. Wind-induced microphone noise detection-automatically monitoring the audio quality of field recordings. In 2013 IEEE International Conference on Multimedia and Expo (ICME) (pp. 1-6). IEEE.
- Nemer, E. and Leblanc, W., 2009, October. Single-microphone wind noise reduction by adaptive postfiltering. In 2009 IEEE Workshop on Applications of Signal Processing to Audio and Acoustics (pp. 177-180). IEEE.
4. The paper present some info about wake effect (lines 597 -603) and includes many figures in the Appendix A showing differences in wind and turbine output over time. There are many papers investigating the wakes produced by wind farm turbines and few examples are:
- Barlas, E., Zhu, W.J., Shen, W.Z., Kelly, M. and Andersen, S.J., 2017. Effects of wind turbine wake on atmospheric sound propagation. Applied Acoustics, 122, pp.51-61.
- Heimann, D., Käsler, Y. and Gross, G., 2011. The wake of a wind turbine and its influence on sound propagation. Meteorologische Zeitschrift, 20(4), pp.449-460.I recommend to remove Appendix A and to give a brief literature review of investigations of turbine wake influence to turbine noise.
5. The paper is taking about SA EPA updated the guidelines in 2009 [20] (line206) but the new update guidelines were published in 2019. Please update the paper also.
6. I did not catch meaning of the sentence “ETSU-R–97 identifies turbine installations used as part of the assessment process were for turbine installations that did not give rise to noise complaints” (lines 75-76) please explain it.
Author Response
The three reviewers presented different interpretations of the manuscript for which we accept the majority of the comments, and thank them for their time in reviewing the document and their frank comments.
As a result of the three reviews the document has been changed to a technical advice in relation to the need to determine the actual noise contribution of a wind farm. The title of the revised manuscript is Determination of Acoustic Compliance of Wind Farms. The technical advice doument discusses the need to remove wind noise, wind generated vegetation noise, address different wind directions, and the impact of turbine wake to use an incorrect wind farm wind data.
The revised document considers four case studies of measurements to address the issues raised. The document does not discuss health or sleep impacts (perceived or otherwise) from residents as a result of operational wind farms.
The revised paper highlights the ramification of the absence of identifying the actual wind farm noise contribution to established criteria and current psychoacoustics research - which was the original intent of the article but got lost in the wider problems of wind turbine noise.
We are of the opinion the revised manuscript highlights an important issue in relation to acoustic compliance that will be of assistance to persons undertaking wind turbine research into sleep and the subjective assessment of wind turbines.
Response to Reviewer 1 Comments
Comment 1.
The paper introduction starts from the sentence about requirement to “take account of community reaction or studies into different noise sources” ( line 26) but only very limited information about investigations of wind farm noise to health effect is presented in the end of the paper (lines 701-705). There are a lot of publications on this matter and review of state of art in the area can be included to the paper. Few publications on this matter:
Response 1.
The revised manuscript no longer needs to respond to this comment as it is not the aim of the article.
Comment 2.
The paper expresses doubts in the applicability of the ETSU-R-97 criteria being applied to current wind farm turbines having a capacity greater than 5 MW (lines 15-18). I recommend to include a description of reasons for these doubts. The applied acoustic criteria do not depend on the wind turbine power from physical acoustic point. I recommend to the authors to include brief description of physical phenomena of wind turbine noise and to try explain why the ETSU-R-97 criteria cannot be used for powerful wind turbines. I like NASA publication of this matter: Hubbard, H.H. and Shepherd, K.P., 2009. Wind turbine acoustics. https://ntrs.nasa.gov/archive/nasa/casi.ntrs.nasa.gov/20090026531.pdf
Response 2.
It is correct that the external acoustic criteria do not depend upon wind turbine power. It is the resultant noise levels are receptor locations do depend upon the wind turbine sound power level. The ETSU-R-97 criteria is purported to be based upon the protection of sleep - but is not based on any sleep studies into wind turbine noise. The fundamental issue with ETSU-R-97 is the provision of a regression analysis method that does not identify the noise contribution of the wind farm and uses an averaged wind speed/noise approach without providing any qualifications of the relationship of the wind farm to the receptor location or the power output of the wind farm. The NASA project for wind turbines (starting with Mod-0) related to single turbines. The ETSU-R-97 document ignores a number of critical issues identified by Hubbard in the above reference - such as directivity of turbines and the absence of predictions taking into account wake turbulence. Kelley and Hubbard's work in relation to modal frequencies and structure resonances in dwellings that can give rise to audible noise, vibration and sensation(primarily from the Mod-1 turbine) are ignored in the majority of noise limits from wind turbines. ETSU-R-97 ignores internal noise levels. From identifying what can give rise to annoyance and complaints inside dwellings I prefer the Kelley report for the Mod-1 turbine
Comment 3.
The main paper result is connected with assessment of influence of noise background and noise on microphones. There are many publications of this matter and state of art in this area can be included to the paper. The results of the paper are needed to be compared with the previous published data.
Response 3.
I disagree with the comment with respect to context of the original version and the revised article. There are publications in relation to wind effects on the microphone diaphram and for wind turbines papers on double wind screens for the assessment of infrasound and low frequency noise from wind turbines to which I have personally discussed those designs and other designs with persons working in the area of monitoring wind farm noise. I was also involved in the final review of ANSIS12.9-2016/part 7. It would seem we are at cross purposes and therefore have not explained the intent of thoses measurements. We were addressing the noise floor of wind at various speeds instrumentation used to unde take ambient measurements and specifically the difference for measurements in proximity to vegetation affected by wind. Your Barlos reference in comment 4 refers to wind vegetation noise.
Two case studies were free of wind turbine noise and reveal in low ambient environments the influence of wind induced vegetation noise requires larger separation distances than suggested by Hansen's recent paper for the special edition on wind turbine noise and Bolin's work referenced in your Barlos reference.
Comment 4.
The paper present some info about wake effect (lines 597 -603) and includes many figures in the Appendix A showing differences in wind and turbine output over time. There are many papers investigating the wakes produced by wind farm turbines and few examples are:
Response 4.
Your comment is correct. However, there is little information on actual operational wind farms. The important aspect of the wake effect is that there can be an overall reduction if the sound power of a wind farm, there can be a focusing effect that give rise to significant increase in noise levels at locations removed from the wind farm and that the wake effect can give rise to significant pulsations.
The important aspect in relation to acoustic compliance testing is the use of wake free wind data that is generally higher than the actual wind across the wind farm. Use of the wake free data results in a higher wind speed and shifts the wind speed on the regression line to the right. This results in a misrepresentation of wind versus noise levels by way of the regression analysis method or even the on-off testing method.
Comment 5.
The paper is taking about SA EPA updated the guidelines in 2009 [20] (line206) but the new update guidelines were published in 2019. Please update the paper also.
Response 5.
in 2019 the SA EPA issued a "draft for consultation" updated guideline (noted in the revised manuscript). The updated guildeline has not been issued. Checking the SA EPA website 5 minutes ago and the guideline is still the 2009 version.
Two important variations in the draft 2019 document is the specification that the wind farm contribution is the regression line less the ambient regression line, where the subtraction is a logarithmic subtraction, and that the compliance method is to be for a downwind wind.
Comment 6.
I did not catch meaning of the sentence “ETSU-R–97 identifies turbine installations used as part of the assessment process were for turbine installations that did not give rise to noise complaints” (lines 75-76) please explain it.
Response 6.
New text in revised manuscript:
Table 5 in the ETSU report identifies 18 wind farms used in a survey of public reaction to noise from wind turbines, (as reported to the environmental health departments as of February 1994) to utilize turbines having a rated power varying from 225 kW to 450 kW.
Table 6 in the ETSU report identifies 13 of the 18 windfarms had summarized the number of complaints. Only 5 windfarms reported noise complaints.
Reviewer 2 Report
The paper deals with an interesting topic about the applicability of ETSU-R-97 criteria to recently installed wind turbines. I have several comments I would like the authors to address before considering the paper further.
- Please try rephrasing the title: not all readers will be familiar with ETSU-R-97, so maybe replace with a more general definition.
- The concept of “modern day” is ambiguous – I would recommend replacing throughout the manuscript with more precise definition – e.g., installed after X date…
- The first part of the introduction (approx. lines 24-47, until “of large aircraft [9].”) is not relevant for this paper? It should definitely be removed. A proper scientific literature review on the topic of assessment methods for wind turbine noise is missing and should be added and discussed.
- The Introduction should discuss potential health and psychological distress implications wind turbine noise can have on residents and discuss why it is crucial to have adequate assessment methods.
- The review is focusing on a “regional” problem. This should be put in a more global context – e.g., comparing this method with other methods used around the world and so on…
- Considering the large use of acronyms, I would add an abbreviation list at the beginning or end of the manuscript.
- Section 4 - Something is wrong with the first sentence of Section 4 – please revise.
- “Experiments” does not look right as title for section 4 – please consider alternatives (e.g., Case Study or alike). This whole section should be reorganized into subsections that describe the methodology – i.e., case study location and description of the site; protocol of measurements; equipment, etc. Then one could have the section about results and discussion of the results.
- In general the reader feels the connection between the first part of the paper (review) and the second part (measurements) is not straightforward, so more efforts should be made in order to make the transition smoother.
- The general quality of the figures is quite low, some of them are not even readable – they should be made consistent in terms of size, visual identity, font type/size, scales, etc. Units should always be reported on axis according to standards, etc. This also applies to the figures in the Appendix.
- The following sentence “A classic picture to show the impact of the wake behind turbines is the Horns Rev Offshore Wind farm in Denmark.” And the corresponding Figure 10 should be removed as it is redundant.
- First sentences of section 6 make no sense, please revise.
- The sentence “In December 2016, the very issue of what constitutes sleep disturbance and adverse impacts as 679 a result of wind farm operations was raised by with the Australian Wind Farm Commissioner to 680 which there has been no response 3 ¼ years later, despite requests by the author and others for 681 answers to the following questions:” and the following questions should be removed as they are not relevant. The same concept could be expressed by rephrasing it into 1 sentence.
- Sections 5,6,7 should be reorganize and merged into a single section “Discussion and Conclusions”
- As a general comment, it is clear to this reviewer that the authors have considerable experience in professional assessment of wind turbine noise. However, the manuscript should be reworked to give it more scientific rigour and credibility. Personal opinions or statements not referenced and/or clearly supported by empirical data reported in the paper should be dismissed.
Author Response
The three reviewers presented different interpretations of the manuscript for which we accept the majority of the comments , and thank them for their time in reviewing the document and their frank comments..
As a result of the three reviews the document has been changed to a technical advice in relation to the need to determine the actual noise contribution of a wind farm. The title of the revised manuscript is Determination of Acoustic Compliance of Wind Farms. The technical advice document discusses the need to remove wind noise, wind generated vegetation noise, address different wind directions, and the impact of turbine wake to use an incorrect wind farm wind data.
The revised document considers four case studies of measurements to address the issues raised. The document does not discuss health or sleep impacts (perceived or otherwise) from residents as a result of operational wind farms.
The revised paper highlights the ramification of the absence of identifying the actual wind farm noise contribution to established criteria and current psychoacoustics research - which was the original intent of the article but got lost in the wider problems of wind turbine noise.
We are of the opinion the revised manuscript highlights an important issue in relation to acoustic compliance that will be of assistance to persons undertaking wind turbine research into sleep and the subjective assessment of wind turbines.
Response to Reviewer 2 Comments
The paper deals with an interesting topic about the applicability of ETSU-R-97 criteria to recently installed wind turbines. I have several comments I would like the authors to address before considering the paper further.
Comment 1.
Please try rephrasing the title: not all readers will be familiar with ETSU-R-97, so maybe replace with a more general definition.
Response 1.
Paper now a technical advice and titled: Determination of Acoustic Compliance of Wind Farms
Comment 2.
The concept of “modern day” is ambiguous – I would recommend replacing throughout the manuscript with more precise definition – e.g., installed after X date…
Response 2.
No longer required in revised manuscript.
Comment 3.
The first part of the introduction (approx. lines 24-47, until “of large aircraft [9].”) is not relevant for this paper? It should definitely be removed. A proper scientific literature review on the topic of assessment methods for wind turbine noise is missing and should be added and discussed.
Response 3.
First part of introduction has been removed – the introduction has been re-written. The change to a technical advice on acoustic compliance no longer requires comments on assessment methods. ETSU-R-97 does not assess wind turbine noise. The topic of compliance testing does not involve any assessment of the noise.
Comment 4.
The Introduction should discuss potential health and psychological distress implications wind turbine noise can have on residents and discuss why it is crucial to have adequate assessment methods.
Response 4.
As the subject is about the technical aspect of determining the actual wind farm contribution then there is no longer a need to discuss potential heal and psychological distress implications. The outcome of obtaining the true wind farm noise contribution could very well change the assessment methods by others.
Comment 5.
The review is focusing on a “regional” problem. This should be put in a more global context – e.g., comparing this method with other methods used around the world and so on…
Response 6.
The comment is noted. The introduction now provides criteria for Europe, Canada and the US. The focus of the technical advice is about the derived noise contribution whether it is assessed as an Leq metric or a regression analysis method. There is a requirement to use case studies which are measurements conducted in Australia and most importantly the Cape Bridgewater Study on which both authors were involved.
Comment 6.
Considering the large use of acronyms, I would add an abbreviation list at the beginning or end of the manuscript.
Response 6.
An abbreviation list in now at the end of the manuscript.
Comment 7.
Section 4 - Something is wrong with the first sentence of Section 4 – please revise.
Response 7.
Revised.
Comment 8.
“Experiments” does not look right as title for section 4 – please consider alternatives (e.g., Case Study or alike). This whole section should be reorganized into subsections that describe the methodology – i.e., case study location and description of the site; protocol of measurements; equipment, etc. Then one could have the section about results and discussion of the results.
Response 8.
This comment was considered to be the most important comment of all the reviewers. As a result the experiments have become Case Studies (in a different order) and the Wake Effect has become a Case Study.
Comment 9.
In general the reader feels the connection between the first part of the paper (review) and the second part (measurements) is not straightforward, so more efforts should be made in order to make the transition smoother.
Response 9.
Addressed in re-write of the manuscript
Comment 10.
The general quality of the figures is quite low, some of them are not even readable – they should be made consistent in terms of size, visual identity, font type/size, scales, etc. Units should always be reported on axis according to standards, etc. This also applies to the figures in the Appendix.
Response 10.
Size of figures has been increased. Figures in Appendix have been removed and referenced in the text.
Comment 11.
The following sentence “A classic picture to show the impact of the wake behind turbines is the Horns Rev Offshore Wind farm in Denmark.” And the corresponding Figure 10 should be removed as it is redundant.
Response 11.
Sentence and picture removed.
Comment 12.
First sentences of section 6 make no sense, please revise.
Response 12.
Revised
Comment 13.
The sentence “In December 2016, the very issue of what constitutes sleep disturbance and adverse impacts as 679 a result of wind farm operations was raised by with the Australian Wind Farm Commissioner to 680 which there has been no response 3 ¼ years later, despite requests by the author and others for 681 answers to the following questions:” and the following questions should be removed as they are not relevant. The same concept could be expressed by rephrasing it into 1 sentence.
Response 13.
Removed from manuscript as no longer relevant in the context of discussion of acoustic compliance and determination of noise contribution. When the contribution is derived then the relevance of sleep disturbance and annoyance impact under current research may change.
Comment 14
Sections 5,6,7 should be reorganize and merged into a single section “Discussion and Conclusions”
Response 14.
On the suggestion of Case Studies, each Case study is subdivided into methodology, results and discussion. Under the revised title there is a new conclusion.
Comment 15
As a general comment, it is clear to this reviewer that the authors have considerable experience in professional assessment of wind turbine noise. However, the manuscript should be reworked to give it more scientific rigour and credibility. Personal opinions or statements not referenced and/or clearly supported by empirical data reported in the paper should be dismissed.
Response 15.
Thank you for the kind comments. The revised manuscript has been re-worked to address this comment.
Reviewer 3 Report
The paper is presented as a review, but it has no connotates for being so. There are many guidelines for performing scientific reviews that I suggest the authors to read. Obviously, the paper cannot be conceived as original paper, but it can be eventually re-worked to be technical note, or can be submitted to local (national audience).
I believe the paper is more suitable for locale audience, because it is not true that ETSU-R-97 is so important around the world, but if the authors still want to follow the actual path of a review, they should really increase the number of international legislations covered, showing where the ETSU-R-97 is really used or considered and where not. Suggestions about this path are reported in derails to the authors. If, otherwise, the authors would explore the technical note way, in this case the focus should be more on measurements, reducing a lot the talks, and showing your analytical results.
- The sentence “The basis of acoustic criteria for wind farms is ETSU-R-97 “The Assessment and Rating of Noise from Wind Farms” in the abstract is a bit assumptive. Especially in the abstract, without the chance for justifying it.
- The first page of the introduction is totally out of lines. There is no need to explore other areas, but it much more important to start with a better introduction of wind turbines, wind turbines noise and their health effects.
- On such regards, a list of possible references is reported next:
Basner, M., Babisch, W., Davis, A., Brink, M., Clark, C., Janssen, S., Stansfeld, S., 2014. Auditory and non-auditory effects of noise on health. Lancet 383 (9925), 1325–1332. https://doi.org/10.1016/S0140-6736(13)61613-X. Berglund, B., Lindvall, T., Schwela, D.H., 1999. Guidelines for Community Noise. World Health Organization. Groothuis-Oudshoorn, C.G., Miedema, H.M., 2006. Multilevel grouped regression for analyzing self-reported health in relation to environmental factors: the model and its application. Biom. J. 48 (1), 67–82. https://doi.org/10.1002/bimj.200410172. Horner, B., Krogh, C., Jeffrey, R., 2013. Audit report: literature reviews on wind turbine noise and health. Proceedings of the 5th International conference on Wind Turbine Noise, Denver. Janssen, S.A., Vos, H., Eisses, A.R., Pedersen, E., 2011. A comparison between exposureresponse relationships for wind turbine annoyance and annoyance due to other noise sources. J. Acoust. Soc. Am. 130 (6), 3743–3756. https://doi.org/10.1121/ 1.3653984. Kaldellis, J.K., Garakis, K., Kapsali, M., 2012. Noise impact assessment on the basis of onsite acoustic noise immission measurements for a representative wind farm. Renew. Energy 41, 306–314. https://doi.org/10.1016/j.renene.2011.11.009. Klæboe, R., 2011. Noise and health: annoyance and interference. In: Nriagu, J.O. (Ed.), Encyclopedia of Environmental Health. Elsevier, Burlington, CA, USA, pp. 152–163. Knopper, L.D., Ollson, C.A., 2011. Health effects and wind turbines: a review of the literature. Environ. Health 10, 78. https://doi.org/10.1186/1476-069X-10-78. Knopper, L.D., Ollson, C.A., McCallum, L.C., Whitfield Aslund, M.L., Berger, R.G., Souweine, K., McDaniel, M., 2014. Wind turbines and human health. Front. Public Health 2, 63. https://doi.org/10.3389/fpubh.2014.00063. Kurpas, D., Mroczek, B., Karakiewicz, B., Kassolik, K., Andrzejewski, W., 2013. Health impact of wind farms. Ann. Agric. Environ. Med. 20 (3), 595–605. Lee, S., Kim, K., Choi, W., Lee, S., 2011. Annoyance caused by amplitude modulation of wind turbine noise. Noise Control Eng. 59 (1). https://doi.org/10.3397/1.3531797 (pp. 38–46(9)). Fredianelli, Luca, Stefano Carpita, and Gaetano Licitra. “A procedure for deriving wind turbine noise limits by taking into account annoyance.” Science of the total environment 648 (2019): 728-736.
- Other legislative framework to be analyzed and include can be those from Germany, France, Netherlands, Canada, Denmark, Italy, …
- When mentioning noise measurements procedure, there are some new deserving attention, such as the Italian Fredianelli, Luca, et al. "Analytical assessment of wind turbine noise impact at receiver by means of residual noise determination without the wind farm shutdown." Noise Control Engineering Journal 65.5 (2017): 417-433.
Author Response
The three reviewers presented different interpretations of the manuscript for which we accept the majority of the comments , and thank them for their time in reviewing the document and their frank comments..
As a result of the three reviews the document has been changed to a technical advice in relation to the need to determine the actual noise contribution of a wind farm. The title of the revised manuscript is Determination of Acoustic Compliance of Wind Farms. The technical advice document discusses the need to remove wind noise, wind generated vegetation noise, address different wind directions, and the impact of turbine wake to use an incorrect wind farm wind data.
The revised document considers four case studies of measurements to address the issues raised. The document does not discuss health or sleep impacts (perceived or otherwise) from residents as a result of operational wind farms.
The revised paper highlights the ramification of the absence of identifying the actual wind farm noise contribution to established criteria and current psychoacoustics research - which was the original intent of the article but got lost in the wider problems of wind turbine noise.
We are of the opinion the revised manuscript highlights an important issue in relation to acoustic compliance that will be of assistance to persons undertaking wind turbine research into sleep and the subjective assessment of wind turbines.
Response to Reviewer 3 Comments
Comment 1.
The paper is presented as a review, but it has no connotates for being so. There are many guidelines for performing scientific reviews that I suggest the authors to read. Obviously, the paper cannot be conceived as original paper, but it can be eventually re-worked to be technical note, or can be submitted to local (national audience).
Response 1.
The manuscript has been amended to be a Technical Advice. If you the document to be callsed a Technical Note we have no objection.
Comment 2.
I believe the paper is more suitable for locale audience, because it is not true that ETSU-R-97 is so important around the world, but if the authors still want to follow the actual path of a review, they should really increase the number of international legislations covered, showing where the ETSU-R-97 is really used or considered and where not. Suggestions about this path are reported in derails to the authors. If, otherwise, the authors would explore the technical note way, in this case the focus should be more on measurements, reducing a lot the talks, and showing your analytical results.
Response 2.
The change in the title is relevant for all the world – whether the acoustic compliance method is by absolute on-off testing, an absolute level, a background + 5 dB(A) level or the regression analysis method. The number of international legislations have been increased and the change to Case Studies has focussed on the measurements.
Comment 3.
The sentence “The basis of acoustic criteria for wind farms is ETSU-R-97 “The Assessment and Rating of Noise from Wind Farms” in the abstract is a bit assumptive. Especially in the abstract, without the chance for justifying it.
Response 3.
Deleted.
Comment 4.
The first page of the introduction is totally out of lines. There is no need to explore other areas, but it much more important to start with a better introduction of wind turbines, wind turbines noise and their health effects.
On such regards, a list of possible references is reported next:
Response 4.
First sentence noted. The second sentence is no longer required as the technical advice is not related to health effects or impacts, or the merit of acoustic criteria. As a result of the change in the title the suggested references are not applicable.
Comment 5.
Other legislative framework to be analyzed and include can be those from Germany, France, Netherlands, Canada, Denmark, Italy, …
Response 5.
Listed but not required to be analysed as a result of the change in topic.
Comment 6.
When mentioning noise measurements procedure, there are some new deserving attention, such as the Italian Fredianelli, Luca, et al. "Analytical assessment of wind turbine noise impact at receiver by means of residual noise determination without the wind farm shutdown." Noise Control Engineering Journal 65.5 (2017): 417-433.
Response 6.
A valid comment for residual noise but what two of the case studies were addressing was wind vegetation noise which relates to references 7 – 9 in the above reference. Fredianelli, Luca et al were seeking to do is remove residual noise insitu which is relevant for undertaking compliance testing without having shutdowns. How to derive the dB(A) contribution without removing the wind induced vegetation noise was not addressed. The revised manuscript is focused on the inability of the current methods to determine the actual wind farm noise contribution. The use of wake free wind data is misleading and cannot be used for identification of the wind farm operations for either on-off testing or regression analysis. Hence in our view it is not a local/regional issues.
Round 2
Reviewer 1 Report
The authors conducted a huge work and the paper has several interesting results that are worth to be published. The paper was significantly improved based on the reviewer's comments.
Author Response
Response to amended manuscript – requesting minor edits
As per the original manuscript the responses to the amended manuscript cover different paths which is of benefit it getting a good manuscript for publication and for which we are grateful for the input from the reviewers.
Reviewer 1 provided comments in relation to the improvements and identified several interesting results that are worth to be published. To which we are thankful.
Reviewer 1 did not provide any comments or requests for modification of the amended manuscript.
Reviewer 2 identified the amended manuscript was considerably improved from the original document.
Reviewer 2 provided 3 comments which have been addressed in the attached document:
Comment 1
The case studies section could still be slightly re-organized so that the structure of the subsections is "symmetric" among case studies: currently each case study has different titles for its subsections. This would greatly improve readability.
Response to Comment 1
We agree with the comment. We have modified the case studies to have similar headings for the subsections.
Comment 2
If some figures (lke Fig 1 or 2) are not the making of the authors copyrights issues should be checked.
Response to Comment 2
We have removed the previous figures 1 & 2. In response to reviewer 3’s request for implications of wind turbine noise and health effect we added in criteria from the WHO 2018 Guidelines for the European Region. This identified the material in Figure 1 was out of date. To this end we found the wind turbine regression curves in the WHO document are not all Lden. So we created our version of the three dose response curves all in Lden as Figure 1. We then changed the noise level axis to Leq to permit a comparison of the Leq limits used around the world as Figure 2.
The previous Figure 2 (referred to in the comment) has been changed to utilise an actual regression curve from the Cape Bridgewater study. In the process we have highlighted the difficulties in getting wind data on the wind farm and the fact that operational regression curves are not 2nd order polynomial curves like the original figure 2 but close to linear or 2nd order polynomial where the x2 term is negative.
Therefore providing our own graphs to replace Figures 1 & 2 has provided additional material that has not been previously identified and we feel is of assistance in the presenting out case.
Comment 3
The quality of the remaining figures has been improved but the visual outlook is still not exactly consistent.
Response to comment 3
We agree with the comment. Part of the issue was different analysis packages providing different graphical presentations. We have modified the graphs to utilise small data symbols, grid lines and standardised the size to address the comment and provide consistency.
Third Reviewer
Comment 1
The authors applied many changes as requested by all the reviewers and the quality of the work has improved.
Response to Comment 1. Thank you
Comment 2
Unfortunately, the authors only sent the document with correction highlighted and it is not easier to fully understand the paper in all its entity. Thus, i will need to read it back again after another round.
Response to Comment 2
As the amended manuscript bore no relationship to the original submission we felt providing a tracked version would be more confusion as sections had been moved around.
For the revision of the amended manuscript we provide a tracked version of the document and a clean version of the document (having accepted the tracking). There are a significant number of amendments as the insertion of the WHO 2018 guidelines earlier in the document and other references required caused all the reference numbering to be changed. The additional figures arising from changing figures 2 & 3, and working splitting some figures resulted in changing the number of figures.
Comment 3
In the meanwhile, i ask the authors to put back some lines with the implications of wind turbine noise and health effect, which, differently from what lastly reported by them, it is always important for the audience.
Response to Comment 3
This request caused some concerns in that covering those two components would really require many more pages and a significant increase in the references.
Our response to this request has been to rely upon the WHO 2018 guideline and annoyance curves for wind turbines and their comments on not enough studies for health/sleep effects in the introduction (lines 73 – 79), Figures 1 & 2 and additional comments in the conclusion.
Reviewer 2 Report
I think the paper is considerably improved.
The case studies section could still be slightly re-organized so that the structure of the subsections is "symmetric" among case studies: currently each case study has different titles for its subsections. This would greatly improve readability.
If some figures (lke Fig 1 or 2) are not the making of the authors copyrights issues should be checked.
The quality of the remaining figures has been improved but the visual outlook is still not exactly consistent.
Thank you for addressing the points of the first round of review.
Author Response
Response to amended manuscript – requesting minor edits
As per the original manuscript the responses to the amended manuscript cover different paths which is of benefit it getting a good manuscript for publication and for which we are grateful for the input from the reviewers.
Comment 1
The case studies section could still be slightly re-organized so that the structure of the subsections is "symmetric" among case studies: currently each case study has different titles for its subsections. This would greatly improve readability.
Response to Comment 1
We agree with the comment. We have modified the case studies to have similar headings for the subsections.
Comment 2
If some figures (lke Fig 1 or 2) are not the making of the authors copyrights issues should be checked.
Response to Comment 2
We have removed the previous figures 1 & 2. In response to reviewer 3’s request for implications of wind turbine noise and health effect we added in criteria from the WHO 2018 Guidelines for the European Region. This identified the material in Figure 1 was out of date. To this end we found the wind turbine regression curves in the WHO document are not all Lden. So we created our version of the three dose response curves all in Lden as Figure 1. We then changed the noise level axis to Leq to permit a comparison of the Leq limits used around the world as Figure 2.
The previous Figure 2 (referred to in the comment) has been changed to utilise an actual regression curve from the Cape Bridgewater study. In the process we have highlighted the difficulties in getting wind data on the wind farm and the fact that operational regression curves are not 2nd order polynomial curves like the original figure 2 but close to linear or 2nd order polynomial where the x2 term is negative.
Therefore providing our own graphs to replace Figures 1 & 2 has provided additional material that has not been previously identified and we feel is of assistance in the presenting out case.
Comment 3
The quality of the remaining figures has been improved but the visual outlook is still not exactly consistent.
Response to comment 3
We agree with the comment. Part of the issue was different analysis packages providing different graphical presentations. We have modified the graphs to utilise small data symbols, grid lines and standardised the size to address the comment and provide consistency.

Reviewer 3 Report
The authors applied many changes as requested by all the reviewers and the quality of the work has improved.
Unfortunately, the authors only sent the document with correction highlighted and it is not easier to fully understand the paper in all its entity. Thus, i will need to read it back again after another round.
In the meanwhile, i ask the authors to put back some lines with the implications of wind turbine noise and health effect, which, differently from what lastly reported by them, it is always important for the audience.
Author Response
Response to amended manuscript – requesting minor edits
As per the original manuscript the responses to the amended manuscript cover different paths which is of benefit it getting a good manuscript for publication and for which we are grateful for the input from the reviewers.
Comment 1
The authors applied many changes as requested by all the reviewers and the quality of the work has improved.
Response to Comment 1. Thank you
Comment 2
Unfortunately, the authors only sent the document with correction highlighted and it is not easier to fully understand the paper in all its entity. Thus, i will need to read it back again after another round.
Response to Comment 2
As the amended manuscript bore no relationship to the original submission we felt providing a tracked version would be more confusion as sections had been moved around.
For the revision of the amended manuscript we provide a tracked version of the document and a clean version of the document (having accepted the tracking). There are a significant number of amendments as the insertion of the WHO 2018 guidelines earlier in the document and other references required caused all the reference numbering to be changed. The additional figures arising from changing figures 2 & 3, and working splitting some figures resulted in changing the number of figures.
Comment 3
In the meanwhile, i ask the authors to put back some lines with the implications of wind turbine noise and health effect, which, differently from what lastly reported by them, it is always important for the audience.
Response to Comment 3
This request caused some concerns in that covering those two components would really require many more pages and a significant increase in the references.
Our response to this request has been to rely upon the WHO 2018 guideline and annoyance curves for wind turbines and their comments on not enough studies for health/sleep effects in the introduction (lines 73 – 79), Figures 1 & 2 and additional comments in the conclusion.

Round 3
Reviewer 3 Report
The paper is fine to be published